# Novel Plant-Associated *Brevibacillus* and *Lysinibacillus* Genomospecies Harbor a Rich Biosynthetic Potential of Antimicrobial Compounds

**DOI:** 10.3390/microorganisms11010168

**Published:** 2023-01-09

**Authors:** Jennifer Jähne, Thanh Tam Le Thi, Christian Blumenscheit, Andy Schneider, Thi Luong Pham, Phuong Thao Le Thi, Jochen Blom, Joachim Vater, Thomas Schweder, Peter Lasch, Rainer Borriss

**Affiliations:** 1Proteomics and Spectroscopy Unit (ZBS6), Center for Biological Threats and Special Pathogens, Robert Koch Institute, 13353 Berlin, Germany; 2Division of Pathology and Phyto-Immunology, Plant Protection Research Institute (PPRI), Duc Thang, Bac Tu Liem, Ha Noi, Vietnam; 3Bioinformatics and Systems Biology, Faculty of Biology and Chemistry, Justus-Liebig Universität Giessen, 35392 Giessen, Germany; 4Institute of Marine Biotechnology e.V. (IMaB), 17489 Greifswald, Germany; 5Pharmaceutical Biotechnology, University of Greifswald, 17489 Greifswald, Germany; 6Institute of Biology, Humboldt University Berlin, 10115 Berlin, Germany

**Keywords:** *Lysinibacillus*, *Brevibacillus*, genomic islands, phylogenomics, taxonomy, ANI, average nucleotide identity, dDDH, DNA-DNA hybridization: biocontrol, nematicidal activity, plant growth promotion, secondary metabolites

## Abstract

We have previously reported the draft genome sequences of 59 endospore-forming Gram-positive bacterial strains isolated from Vietnamese crop plants due to their ability to suppress plant pathogens. Based on their draft genome sequence, eleven of them were assigned to the *Brevibacillus* and one to the *Lysinibacillus* genus. Further analysis including full genome sequencing revealed that several of these strains represent novel genomospecies. In vitro and in vivo assays demonstrated their ability to promote plant growth, as well as the strong biocontrol potential of *Brevibacilli* directed against phytopathogenic bacteria, fungi, and nematodes. Genome mining identified 157 natural product biosynthesis gene clusters (BGCs), including 36 novel BGCs not present in the MIBiG data bank. Our findings indicate that plant-associated *Brevibacilli* are a rich source of putative antimicrobial compounds and might serve as a valuable starting point for the development of novel biocontrol agents.

## 1. Introduction

Coffee and pepper are valuable products of Vietnamese agriculture. Their coffee production worldwide ranks second, after Brazil. Black pepper is of similar importance in Vietnam, and has been exported to more than 110 countries. However, during the last years, there has been a tendency of harvest yield reduction in the case of coffee of about 20% compared to the yield of 2014. Harvest losses are caused by plant pathogens such as fungi (e.g., *Fusarium oxysporum*), oomycetes (*Phytophthora palmivora*) and root-knot nematodes (e.g., *Meloidogyne incognita*). In the past, chemical pesticides were used to control the phytopathogens, but their use is no longer permitted due to their toxic remnants. The epidemic occurrence of fast death disease, which is damaging black pepper plantations, has led to a rural exodus of farmers. For this reason, the sustainable development of agriculture, which includes the applying of highly efficient and reliable biocontrol agents, useful for preventing and suppressing pests on black pepper and coffee, is an urgent need. 

We recently isolated endospore-forming Gram-positive bacteria strains with putative biocontrol functions from healthy Vietnamese crop plants (e.g., coffee and black pepper) grown in fields infested with plant pathogens [1]. Based on their draft genome sequences, 47 isolates were assigned as being representative of the genus *Bacillus* belonging either to the *B. subtilis* or the *B. cereus* group [2]. Twelve additional isolates were either representatives of the *Brevibacillus* (11) or the *Lysinibacillus* (1) genus [1]. The majority of these strains could not be assigned down to species level. Preliminary experiments revealed that the *Brevibacillus* strains exert a strong nematicidal activity, and were able to suppress the growth of fungal plant pathogens. 

The genus *Brevibacillus* encompassing the historical *Bacillus brevis* cluster [3] belongs to the family Bacillaceae of the phylum Firmicutes. The genus currently comprises 30 validated species according to the List of Prokaryotic Names with Standing in Nomenclature (LPSN; [4]). The members of the genus are characterized by the formation of spherical swollen sporangia, containing L-lysine in their peptidoglycan cell wall, and by their inability to utilize traditional carbohydrates. Although reports about the biocontrol activity of *Brevibacilli* against plant pathogens are rare [5,6,7], the application of antimicrobial compounds with medical importance produced by *Brevibacillus* spp. has a long tradition. Members of the *Brevibacillus brevis* cluster are a rich source of antimicrobial peptides and lipopeptides (AMPs), including gramicidin A and gramicidin S. The cyclic decapeptide gramicidin S (Soviet gramicidin) was discovered as early as 1942 [8]. Linear gramicidin A is the first AMP used clinically, and has still medical importance in topical medications. Until today, many novel AMPs such as the non-ribosomally synthesized cyclic decapeptide tyrocidine, the linear lipo-tri-decapeptide brevibacillin, the atypical cationic peptide edeine, and the bacteriocins laterosporulin and laterosporulin10 have been described [8]. *Brevibacillus* Leaf182 has previously been characterized as the most powerful inhibitor in the *Agrostemma thaliana* phyllosphere, and harbors a large number of gene clusters putatively involved in the synthesis of AMPs and other antimicrobial compounds. Marthiapeptide A and the previously unknown polyketide macrobrevin were identified in Leaf182 [9]. 

The genus *Lysinibacillus* [10], with the type species *Lysinibacillus boronitolerans*, at present comprises 21 validly published species (24-10-2022 LPSN). *L. sphaericus*, formerly *Bacillus sphaericus*, is known as an entomopathogen able to act as a biological insecticide that is efficient against mosquitos. In addition, other beneficial abilities such as plant growth promotion, the biocontrol of plant pathogens, and bioremediation have been reported [11].

In this study, we have fully characterized a new *Lysinibacillus* isolate and three novel *Brevibacillus* strains which were isolated from Vietnamese crop plants. We could assign their genomes to novel genomospecies, and were able to identify more than 150 biosynthetic gene clusters (BGCs) probably involved in the synthesis of a multitude of structurally diverse antimicrobial compounds. Biocontrol experiments performed with phytopathogenic bacteria, fungi, and nematodes revealed the high potential of the *Brevibacilli* isolates to suppress plant pathogens. 

## 2. Materials and Methods

### 2.1. Strain Isolation, Growth Conditions and DNA Isolation

The plant-associated bacteria were isolated from healthy crop plants, such as coffee and pepper, from fields located at Dak Nong, Dak Lac provinces, Vietnam, and infested with plant pathogens such as *Phytophthora palmivora*, *Fusarium oxysporum*, and nematode *Meloidogyne* sp. [1]. Routinely, the leaf, stem and root of healthy plants were selected and taken to the laboratory for further processing. The plant organs were cut into 5 × 5 mm pieces and washed twice with sterile water. Afterwards, the plant parts were dipped into 75% ethanol for one min, and then into 0.1% mercury dichloride (HgCl_2_) for two min. The cuts were then washed three times with sterile water, and taken up in 10 mL sterile water. The suspension was grounded in a sterile and chilled mortar. After 30 min. incubation, 0.1 mL of the solution was transferred to LB plates and allowed to grow for 72 h at 28 °C. Finally, single colonies were purified, transferred to 50 mL fresh LB medium, and cultured in a shaker at 28 °C and 180 rpm for 24 h. Due to their slow growth, *Brevibacillus* spp. were found to be enriched when soil samples adherent to plant roots were incubated under shaking for two weeks, diluted to 10^−9^, and then plated either onto R2A agar (Oxoid Lim., Basingstoke, UK) or onto 1/5 diluted NPT agar consisting of 0.4 g/L nutrient broth, 1 g/L potato dextrose, 1.2 g/L tryptic soy broth, 2 g/L MES hydrate (2-morpholino-ethan sulfonic acid), and 15 g/L agar, pH 5.75. Colonies appeared after three to seven days incubation at 28 °C. The purified strains were maintained as glycerol stocks (20 %, *w*/*v*) at −80 °C. The cultivating of the bacterial strains and DNA isolation have been described previously [1,2].

### 2.2. Genome Sequencing, Assembly and Annotation

The genome sequences of *Lysinibacillus* sp. CD3-6, *Brevibacillus parabrevis* HD3.3A, *Brevibacillus* sp. DP1.3A, and *Brevibacillus* sp. M2.1A were reconstructed using a combined approach of two sequencing technologies, which generated short paired-end reads and long reads. The resulting sequences were then used for hybrid assembly. Short-read sequencing was conducted in LGC Genomics (Berlin, Germany) using an Illumina HiSeqq in a paired 150 bp manner, as described previously [2]. Long read sequencing was done in house with the Oxford Nanopore MinION with the Flow Cell (R9.4.1) and prepared with the Ligation Sequencing Kit (SQK-LSK109). The samples were sequenced over 48 h and base called afterwards by Guppy v3.1.5. Long reads were trimmed using Porechop (https://github.com/rrwick/Porechop, v0.2.4, accessed on 3 December 2022) and filtered using Filtlong (https://github.com/rrwick/Filtlong, v0.2.0, accessed on 3 December 2022) on default settings. De-novo assemblies were generated by using the hybrid-assembler Unicycler (https://github.com/rrwick/Unicycler, v0.4.8, accessed on 3 December 2022) [11]. The quality of assemblies was assessed by determining the ratio of falsely trimmed protein by using Ideel (https://github.com/phiweger/ideel, accessed on 3 December 2022). Genome coverage of the obtained contigs was 50× on average. 

Automatic genome annotation was performed using the RAST (Rapid A using Subsystem Technology) server [12,13] implemented with RASTk [14], and with the NCBI Genome Automatic Annotation Pipeline (PGAP6.2, [7]) for the general genome annotation provided by NCBI RefSeq. Functional annotation was done by using PATRIC web resources [15]. Core- and pan-genome analysis was performed within the EDGAR3.0 pipeline [16]. Genomic islands (GI) were predicted with the webserver IslandViewer 4 (http://www.pathogenomics.sfu.ca/islandviewer/, accessed on 3 December 2022) [17]. Circular plots of genome and plasmid sequences were visualized with BioCircos [18].

For assessment of genome similarity and phylogeny, the genome sequence data were uploaded to the Type (Strain) Genome Server (TYGS), available at https://tygs.dsmz.de, accessed on 3 December 2022 [19]. Information on nomenclature was provided by the List of Prokaryotic names with Standing in Nomenclature (LPSN, available at https://lpsn.dsmz.de, accessed on 3 December 2022) [20]. All user genomes were compared with all type strain genomes available in the TYGS database via the MASH algorithm [21], and the ten strains with the smallest MASH distances were chosen per user genome. Using the Genome BLAST Distance Phylogeny approach (GBDP), the ten closest type strain genomes for each of the user genomes were calculated. GTDB sp. were calculated with GTDB-Tk and the FASTANI calculator (https://gtdb.ecogenomic.org/, accessed on 3 December 2022). GTDB-Tk is a toolkit for assigning objective classifications to bacterial and archaeal genomes based on the Genome Database Taxonomy (GTDB). In silico DNA-DNA hybridization (dDDH) values were calculated in the TYGS platform using formula d_4_, which is the sum of all identities found in the high score segment pairs (HSPs) divided by the total length of all HSPs. A pan-genome analysis was performed using the Bacteria Pan Genome Analysis pipeline (BPGA) [22] with the amino acid sequences and default parameter settings (50% identity). The tree files provided by the BPGA pipeline and TYGS were visualized with iTOL (https://itol.embl.de/#, accessed on 3 December 2022).

In addition, the EDGAR3.0 pipeline [16] was used for elucidating taxonomic relationships based on genome sequences. To construct a phylogenetic tree for a project, the core genes of these genomes were computed. In a following step, the alignments of each core gene set are generated using MUSCLE, and the alignments are concatenated to one huge alignment. This alignment is the input for the FastTree software (http://www.microbesonline.org/fasttree/, accessed on 3 December 2022) to generate approximately-maximum-likelihood phylogenetic trees. The values at the branches of FastTree trees are _NOT_ bootstrapping values, but local support values computed by FastTree using the Shimodaira-Hasegawa test. FastANI [23] was used to calculate ANI heatmaps within the EDGAR pipeline for a selected set of genomes.

### 2.3. Biocontrol Activity against Plant Pathogens and Plant Growth Promotion

Antibacterial activity was examined as follows: 0.5 mL of a stationary culture (around 10^9^ cells) of the phytopathogenic indicator bacteria (*Clavibacter michiganensis*, *Dickeya solani*, *Erwinia amylovora*) were mixed with 2 mL liquid soft agar (0.7%) at 40 °C, and then added to Petri dishes with 1.5% LB-Agar. The test bacteria (*Brevibacillus* and *Lysinibacillus* strains) were grown in liquid culture for 48 h. under continuous shaking. 10 µL of the culture was allowed to soak into filter paper discs (2 mm in diameter), and were then placed onto the surface of the soft agar containing the indicator bacteria. The cultures were examined every day and allowed to grow for six days at 27 °C. 

The antifungal activity of the isolates was determined as follows: Plugs (5 mm in diameter) with the pathogenic fungi were placed onto potato dextrose agar (PDA). Paper discs with the test bacteria were then added 20 mm away from the fungi. The cultures were incubated for six days at 27 °C and examined daily. 

A bioassay of nematicidal activity was performed with *Caenorhabditis elegans* N2 (Carolina, U.S.A., https://www.carolina.com, accessed on 3 December 2022) fed with *Escherichia coli* OP50 cells. The culture and synchronization of the worms was performed as previously described [24]. The L4 stage was used for two different bioassays performed as described previously [25]. In the slow killing assay, around 40 L4 *C. elegans* worms were added to the nematode growth medium (NGM) agar plate containing the test bacteria. The mixture was incubated for 3–5 days at 25 °C and inspected daily. In the liquid fast killing assay, the test bacteria were grown overnight under shaking (200 rpm) at 37 °C in 3 mL liquid assay medium. 100 µL of the bacterial culture was diluted with 500 µL M9 medium, and transferred into 12 well plates. Each well was seeded with 40–60 L4 stage N2 nematodes, and the assay was performed at 25 °C for 24 h. Mortalities of nematodes were defined as the ratio of dead nematodes over the tested nematodes. 

Root-knot nematode *Meloidogyne* sp. was isolated from roots of infested pepper plants according to [26]. Tomato plantlets were grown in pots with natural soil under controlled conditions in a greenhouse. Test bacteria and second stage juvenile (J2) nematodes were added to the pots two weeks after transplanting. The number of knots in tomato plants was estimated ten weeks after infestation with the nematodes [27].

A plant growth promotion assay was performed with wild type *Arabidopsis thaliana* (EDVOTEK, USA https://www.edvotek.com/, accessed on 3 December 2022) according to [28]. The surface sterilized seeds were pre-germinated on Petri dishes containing half-strength Murashige-Skoog medium semi-solidified with 0.6% agar and incubated at 22 °C under long daylight conditions (16 h light/8 h dark) for seven days. The roots of *Arabidopsis* seedlings were then dipped into a diluted spore suspension of the test bacteria (10^5^ CFU/mL) for five min., and five seedlings were transferred into a square Petri dish containing half-strength MS-medium solidified with 1% agar. The square Petri dishes were incubated in a growth chamber at 22 °C at a daily photoperiod of 14 h. The fresh weight of the plants was measured 21 days after transplanting for estimation of the ability of bacterial strains for growth promotion. All experiments were performed in triplicate and the standard deviation SD was indicated as bars of the column diagrams.

### 2.4. Identification of Gene Clusters Involved in the Synthesis of Secondary Metabolites

Gene clusters for secondary metabolite synthesis were mined using antiSMASH pipeline version 6 [29] under settings of all features and BAGEL4 [30].

### 2.5. Data Analysis

The data obtained from biocontrol and plant growth promotion experiments were analyzed by Statistic Analysis Systems (SAS) software. The results of three replicates (n = 3) were expressed as SD (standard deviations). Significance was calculated with *t*-tests using the ANOVA procedure according to Duncan at LSD = 0.05. Every experiment was conducted using a completely random design. Excel software was used to create the graphical representations.

### 2.6. Gene Bank Accession Numbers of DNA Sequences

*Lysinibacillus* sp. CD3-6 chromosome: CP085880.1, *Lysinibacillus* sp. CD3-6 extra DNA (2485 bp): CP085881.1, *Brevibacillus parabrevis* HD3.3A chromosome: CP085874.1, *Brevibacillus parabrevis* HD3.3A plasmid (61,606 bp): CP085875.1, *Brevibacillus* sp. DP1.3A chromosome: CP085876.1, *Brevibacillus* sp. DP1.3A plasmid 1 (44,610 bp): CP085877.1, *Brevibacillus* sp. DP1.3A plasmid 2 (10,996 bp): CP085878.1, *Brevibacillus* sp. DP1.3A plasmid 3 (6349 bp): CP085879.1, *Brevibacillus* sp. M2.1A chromosome: NZ_JABSUY020000001.1, NZ_JABSUY020000002.1, NZ_JABSUY020000004.1, *Brevibacillus* sp. M2.1A plasmid (19,434 bp): NZ_JABSUY020000003.1.

## 3. Results and Discussion

### 3.1. Resequencing of Selected Lysinibacillus and Brevibacillus Strains Revealed the Presence of Genomic Islands and Extrachromosomal Elements 

*Lysinibacillus* sp. CD3-6, *Brevibacillus parabrevis* HD3.3A, *Brevibacillus* sp. DP1.3A, and *Brevibacillus* sp. M2.1A were sequenced using nanopore sequencing technology. The complete genome of *Lysinibacillus* sp. CD3-6 consisted of two DNA elements: a single circular chromosome with 4,810,966 bps (CP085880.1), and a small DNA with 2485 bps (CP085881.1). The total size of both DNA elements was 4,813,451 bps harboring 4833 coding sequences. No genes with similarity to plasmid replication proteins were detected in the small DNA element, excluding its definition as a plasmid. A circular plot of the CD3-6 chromosome (4,810,966 bps) computed against the most related genome (*Lysinibacillus* sp. JNUCC-52) is shown in Appendix A. CD3-6 was used as reference for computing the core genome against a set of 15 *Lysinibacillus* genomes representing the most related taxonomic groups (clusters A1-4, see Section 3.2). The core genome consisted of 2733 coding sequences (CDS). The pan genome consisted of 9100 CDS. A total of 187 singletons were detected in CD3-6 when compared with the other 15 genomes (Table 1). Genomic island (GI) prediction using the IslandViewer 4 webserver [17] revealed that the CD3-6 genome was rich in putative genomic islands, mainly characterized by the presence of site-specific integrases, HNH endonucleases, and phage proteins (Figure 1A). GI 1 (618,752–660,952) contained 31 genes including *tnpA* (IS200/IS605 family transposase) and a tyrosine-type recombinase/integrase encoding gene. GI 4 (1,776,323–1,805,601) contained 39 genes including several phage proteins and a gene encoding FAD dependent thymidylate synthase. A predicted class-ii lassopeptide was detected using the antiSMASH pipeline version 6 [29] in GI 5. The largest GI predicted in CD3-6 was GI 8 (2,649,327–2,738,124) harboring 119 genes including the type II toxin-antitoxin system (UED78437.1, UED78438.1) encoded by *hicA* and *hicB*. HicAB modules appear to be highly prone to horizontal gene transfer genes [31]. A complete list of the genes predicted in the CD3-6 GIs is given in Appendix A. 

The chromosome of the isolate *B. parabrevis* HD3.3A (6,154,192 bp) is presented in Appendix A. HD3.3A was used as reference for computing the core genome against representatives of the *Brevibacillus*-A5 branch (clusters 25–27, see Section 3.2). The core genome consisted of 3044 CDS. The pan-genome consisted of 9438 CDS. A total of 166 singletons were detected in HD3.3A (Table 1).

Some 16 GIs were found to be distributed within the HD3.3A chromosome (Figure 1B). They were enriched with transposases of different families, such as IS3, IS21, IS110, IS256, Mu, TnsA, TnsD, Tns7, and site-specific integrases probably involved in horizontal gene transfer. An *agrB* gene probably involved in processing of cyclic lactone autoinducer (AIP) was detected in GI 5. Phage genes were detected in several GIs of this strain. GI 15 (5,552,099–5,579,031) harbored a number of flagellar proteins, possibly affecting the motility behavior of HD3.3A. (Appendix A).

*B. parabrevis* HD3.3A isolate harbored an 61 kb extrachromosomal element (GC:47.3%) containing the gene for the plasmid segregation protein ParM (WP_173600040.1) with similarity (41% identity) to the ParM protein (QYY44717.1) from *Aneurinibacillus thermoaerophilus* plasmid pAT1, and ParM proteins common in the *Bacillus cereus* group [32]. In addition, replicative DnaB helicase (WP_173600059.1), site-specific integrase (WP_229050018.1), and several phage proteins were detected (Figure 2A).

The *Brevibacillus* sp. DP1.3A chromosome consisted of 6601 kb and displayed high similarity with *Brevibacillus* Leaf182 (Appendix A). For estimating the core and pan genome, DP1.3A was computed against representatives of the *Brevibacillus*-A6 branch (clusters 28–35, see Section 3.2). The core genome consisted of 2607 CDs. The pan-genome consisted of 15,934 CDs. A total of 365 singletons were detected in DP1.3A (Table 1). Fourteen GIs were predicted in the chromosome of DP1.3A (Figure 1C). Three GIs contained gene clusters probably involved in the synthesis of secondary metabolites. GI 1 (50,050–55,329) harbored two genes involved in the synthesis of lanthipeptide class-ii peptides. GI 3 (805,782–878,612) harbored a gene cluster probably involved in the synthesis of macrobrevin [9]. Genes involved in the synthesis of a cyclic lactone autoinducer peptide (TIGR04223) were detected in GI 6 (2,311,687–2,324,264). An unknown NRP-PK hybrid scaffold was probably synthesized by genes present in GI 9 (3,310,924–3,397,894). In addition, GI 9 harbored a complete type I DNA restriction-modification system (Appendix A).

*Brevibacillus* sp. DP1.3A harbored three extrachromosomal elements (ECEs). The circular 44 kb DNA element (GC content: 43.9%, Figure 2B) displayed similarity with the HD3.3A plasmid, and also contained the *parM* gene (67.5% identity). In total, both *parM* containing plasmids shared 13 CDS including site-specific integrase, and dUTP diphosphatase. The circular 11 kb DP1.3A plasmid DNA (Figure 2C) encodes the replication initiator protein Rep (WP_173621461.1) involved in DNA-binding and the rolling circle replication of high-copy plasmids [33]. Notably, WP_173621461.1 shared high similarity (88.43% identity) with the Rep63 protein (HAJ4019592.1) from the *E. coli* isolate LA106_16-0310. Present in this plasmid was also a gene encoding the MobA/MobL mobilization protein UED78114.1, which is essential for conjugative plasmid transfer [34]. The protein resembled (74.7%% identity) the MobA/MobL family protein of the gamma proteobacterium *Xanthomonas arbicola* (WP_080960787.1). A third ECE, 6349 bp in size, harbored a similar Rep protein gene as detected in plasmid 2 (Figure 2D).

The *Brevibacillus* sp. M2.1A chromosome (6300 kb) was found to be closely related to the *Brevibacillus brevis* strain 12B3 (Appendix A). Computing with the other members of the *Brevibacillus*-A6 branch yielded 3295 core genes. The pan genome was formed by 13,428 genes. The M2.1A genome contained 113 singletons (Table 1). In contrast with DP1.3A, the GIs detected in the M2.1A chromosome (Figure 1D) did not harbor genes involved in the synthesis of secondary metabolites. A complete thioredoxin system with thioredoxin, thioredoxin-disulfide reductase, and thiol peroxidase was detected in GI 13 (Appendix A). This system might enable M2.1A to respond efficiently against oxidative stress [35].

M2.1A harbored low-copy plasmid DNA (size: 19,434 bps, GC content: 42.0%), whose partition might be governed by the plasmid partition protein A [36] (Table 1). The ParA family protein (MCC8438707.1) shared the highest similarity with the AAA family ATPase from *Brevibacillus borstelensis* (WP_251238174.1, 93.49% identity). In addition, the plasmid also shared partial sequence similarity (16% of their total length) with the low-copy plasmid from *Brevibacillus* sp. DP1.3A (CP085878).

**Table 1 microorganisms-11-00168-t001:** Extrachromosomal elements and general genomic features of *Lysinibacillus* sp. CD3-6, *B. parabrevis* HD3.3A, *Brevibacillus* sp. DP1.3A, and *Brevibacillus* sp. M2.1A. Methods used for generating the data are set in brackets (PGAP = RefSeq, EDGAR). The origin of replication (oriC) was estimated with Ori-Finder 2022 (http://tubic.tju.edu.cn/Ori-Finder2/, accessed on 3 December 2022) [37]. Calculation of core and pan genomes was performed as described in the text (Section 3.1). The protein features estimated with RAST are presented in Appendix A.

Genus	*Lysini-bacillus*	*Brevibacillus*		
Strain	CD3-6	HD3.3A	DP1.3A	M2.1A
*Extrachrosomal elements (ECE)*				
	CP085881 2485 bp	CP085875 61,606 bp ParM-like segregation	CP085877 44,610 bp ParM-like segregation	JABSUY020000003 19,434 bp ParA
			CP085878 10,996 bp Rep, RC replication, MobA/L	
			CP085879 6349 bp	
*Genomic features*				
chromosome	CP085880	CP085874	CP085876	JABSUY020000001
Genome size (bp)	4,810,966	6,154,192	6,601,295	6,172,625
Replication origin (oriC)	4,810,413…4,810,966	6,153,310…6,154,192	6,600,712…6,601,295	570,707…571,289
G+C %	37.1	52.1	47.4	47.4
Number of genes (PGAP)	4794	5819	6203	5956
Genes coding (PGAP)	4589	5570	5960	5728
CDSs total (PGAP)	5331	5661	6030	5783
CDS core genome (EDGAR)	2733	3044	2607	3295
CDS pan genome (EDGAR)	9100	9438	15,934	13,428
CDS singletons (EDGAR)	187	166	365	113
Number of RNAs (PGAP)	149	173	168	166
rRNAs (5S, 16S, 23S, PGAP)	37	38	41	41
tRNAs (PGAP)	112	130	127	106
ncRNAs (PGAP)	5	5	5	5
Pseudo genes (PGAP)	51	76	70	55

### 3.2. Taxonomic Evaluation of Lysinibacillus and Brevibacillus Strains Revealed Novel Genomospecies

#### 3.2.1. Lysinibacillus CD3-6 Forms a Distinct Genomospecies Together with *Lysinibacillus* JNUCC-52

The 16S rRNA gene sequence was extracted using the TYGS server from the whole genome sequence of CD3-6 and used for phylogenetic analysis. We have also directly sequenced the 16S rRNA of CD3-6, which was deposited in the NCBI data base as MW820197.1. Both sequences differ by only one nucleotide. The resulting tree indicated a single species cluster formed by CD3-6 with *Lysinibacillus sphaericus* as the closest related species (Appendix A). In order to exclude the possibility that the type of strains without known genome sequences, but more related to CD3-6, escaped our analysis, we performed a BLASTN-supported search (https://blast.ncbi.nlm.nih.gov, accessed on 3 December 2022) for related 16S rRNA sequences. However, no 16S rDNA sequences with more similarity than *L. sphaericus* (99.61%) were detected.

Since16S rRNA sequences are often not sufficient for species discrimination, we used the genome sequences for taxonomical strain identification. We started our analysis with the CD3-6 genome (CP085880) and 112 *Lysinibacillus* genomes obtained from the NCBI data bank. The phylogenetic tree (Figure 3) suggested that the members of the *Lysinibacillus* genus can be divided into two major groups, A and B. Group B members were not always representatives of the *Lysinibacillus* genus, but were often classified as representatives of different *Ureibacillus* species. Thus, group B seemed to be heterogenous, and to contain different genera. Group A contained *Lysinibacillus* genomes related to the type strain *Lysinibacillus sphaericus* DSM 28 [10], formerly *Bacillus sphaericus*. Based on the cut-off values for species delineation using ANI (96%) and dDDH (70%) [38], 24 clusters (A1–A24) were distinguished (Appendix A). Ten of the clusters contained type strains of recognized *Lysinibacillus* species, but most of the clusters (14) did not contain type strains, and represented according to the definition given by EZBioCloud (https://help.ezbiocloud.net/genomospecies/, accessed on 3 December 2022) for unnamed genomospecies consisting of so far unclassified or wrongly labelled *Lysinibacillus* genomes.

The type strain *Lysinibacillus sphaericus* DSM 28 was classified as being a member of cluster A4. Due to its genome sequence, *Lysinibacillus* sp. CD3-6 was assigned to belong to cluster A2 when using a 70% dDDH radius around each of the 13 *Lysinibacillus* type strains (Figure 3B).

The type strains most related to CD3-6, *Lysinibacillus sphaericus* KCTCC 3346, *Lysinibacillus tabacifolii* K3514, and *Lysinibacillus mangiferihumi*, possessed dDDH values (d4) far below of the species cut off (dDDH < 70, Appendix A). We conclude that CD3-6 represents, together with *Lysinibacillus* JNUCC-52, a novel genomospecies distinguished from the *L. sphaericus* species cluster. It should be noted that *L. mangiferihumi*, *L. tabacifolii*, and *L. varians* were recently characterized as later heterotrophic synonyms of *L. sphaericus* [39], and do not represent valid species.

The extended analysis of whole-genome similarity by their Average Nucleotide Identity (ANI) corroborated the division of the *Lysinibacillus* group A into 24 species clusters (Appendix A). Except clusters A21 and A22, all remaining clusters could be assigned to the genomospecies described in the genome taxonomy database (GTDB) release 07-RS207 (8 April 2022, [40]. The clusterA2 with *Lysinibacillus* CD3-6 together with *Lysinibacillus* JNUCC-52 was assigned as being *Lysinibacillus* sp002340205 following the GTDB taxonomy [41]. Average nucleotide identity and aligned nucleotides [%] were determined using JSpecies (https://jspecies.ribohost.com/, accessed on 3 December 2022) [42]. The method was based on a BLASTN comparison of the genome sequences [43].

#### 3.2.2. Novel GS Were Assigned for Plant-Associated *Brevibacillus* Isolates

The phylogenomic tree constructed with 134 *Brevibacillus* genomes, including the eleven *Brevibacillus* isolates obtained from Vietnamese crop plants, demonstrated that three different major clusters (A, B, C) can be distinguished (Figure 4). Group B consisted of *Brevibacillus laterosporus* strains, whilst group A was formed by representatives of *Brevibacillus brevis* and their closest relatives. Within group A, six major branches with a total of 23 clusters were distinguished. Only 14 of those clusters were covered by type strains, whilst nine clusters represent novel genomospecies without validly recognized type strains. The majority of strains obtained from Vietnamese crop plants clustered within the *Brevibacillus* A group at branch 6. Only the putative *B. parabrevis* strains HD3.3A and HD1.4A clustered within branch 5 (Appendix A). A third cluster, group C, appeared as a heterogenous group of Brevibacilli, displaying a distant relationship to each other and to the other representatives of the *Brevibacillus* taxon (Figure 4A).

According to the analysis routine performed by the Type (Strain) Genome Server (TYGS), five of the Vietnamese *Brevibacillus* isolates clustered together with valid recognized type strains. The *Brevibacillus* strains HD1.4A, and HD3.3A were assigned as *B. parabrevis*, and HB1.1, HB1.2, HB1.4B formed a cluster with the *B. porteri* type strain. The remaining six isolates did not cluster together with validly published type strains and most likely represent four novel GS. *Brevibacillus* sp. RS1.1, HB2.2, and HB1.3 clustered together within one putative novel species cluster, but split into two different subspecies. Two distinguished novel species cluster were formed by *Brevibacillus* sp.: DP1.3A and M2.1A. Another novel GS was formed by *Brevibacillus* MS2.2 together with *Brevibacillus* sp. Leaf182, isolated from *Arabidopsis* phyllosphere [9] (Figure 4A).

The ANI was proposed to replace classical DNA-DNA hybridization (DDH) as the method for prokaryotic species circumscription in 2009 [45]. The FastANI heatmap constructed with the *Brevibacillus* group A genomes (Appendix A) corroborated that the clusters presented in the phylogenomic tree (Appendix A) correctly reflected their taxonomic relationship down to species level when using the cut-off level (95–96%) recommended for interspecies identity [43]. In addition, we have aligned our data with the most recent release of the Genome Taxonomy Database (GTDB) [40] (Release 07-RS207 (8 April 2022), and also found appropriate designations used in Appendix A in case of clusters not covered by recognized type strains. According to the classification given by ANI, dDDH, and GTDB, the query *Brevibacillus* isolate genomes were clustered as follows:Strains HD1.4A and HD3.3A represented the *Brevibacillus parabrevis* species cluster (GS A5-25). They only shared ANIb values of ≤85% with the other group A *Brevibacillus* clusters.The *Brevibacillus porteri* species cluster (GS A6-31) was formed by HB1.1, HB 1.2, and HB1.4B. Their ANIb-values (92–93%), when compared with the other “*brevis*” group strains, were below the species cut-off level (95–96%) recommended as the ANI criterion for interspecies identity.The same was true for HB1.3, HB2.2, and RS1.1 forming together with other genomes GS A6-33. The cluster was designated according to the GTDB classification, as *Brevibacillus brevis* D and did not contain a type strain.Strain MS2.2 formed together with Leaf182 the GS A6-29, designated as *Brevibacillus brevis* C. Their ANI values were found below 93% when compared with the most related clusters of the A6-branch.*Brevibacillus* sp. DP1.3A shared a common cluster (A6-30) with the genome of *Brevibacillus* sp. BC25. The GTDB classification of this cluster was s_*Brevibacillus* sp. 000282075.

*Brevibacillus* sp. M2.1A formed the unique GS A6-34 as a single strain, which was found to be most related to the *Brevibacillus formosus* cluster (GS A6-35). However, ANIb values estimated when compared with this cluster were found to be below the species cut-off (<96%). The GTDB classification of this cluster was s_*Brevibacillus* sp. 013284355.

The strains including their corresponding GS are summarized in Appendix A, and were used for bioassays of their ability to control plant pathogens and to promote plant growth, as described in the next section.

### 3.3. Plant-Associated Brevibacilli Promote Plant Growth and Suppress Plant Pathogens

Our in vitro bioassays demonstrated that the *Brevibacillus* strains did efficiently inhibit the growth of phytopathogenic bacteria, fungi, and nematodes (Appendix A).

#### 3.3.1. Antagonistic Activity against Gram-Positive and Gram-Negative Bacteria

Antagonistic activity against bacteria was indicated by inhibition zones around filter discs containing the test bacteria. The filter discs were placed onto soft agar mixed with the Gram-positive bacterium *Clavibacter michiganensis*, the causative agent of ring-rot in potato tubers. Antibiosis was demonstrated with all tested representatives of the *Brevibacillus* taxon independent of whether cells or supernatants were used. Inhibition zones appeared after one day and increased steadily during the whole period of observation. *Brevibacillus* sp. HB2.2, *Brevibacillus* sp. RS1.1, *B. porteri* HB1.4B, and *Brevibacillus* sp. DP1.3A were among the strains with the highest antagonistic activity against *Clavibacter michiganensis*. However, the antagonistic activity of *Lysinibacillus* CD3-6 was hard to detect (Figure 5).

*Xanthomonas campestris*, the causative agent of bacterial leaf spot disease in pepper, was used as an indicator of antagonistic activity. The largest growth inhibition showed *B. porteri* HB1.1, *Brevibacillus* DP1.3A, and *Brevibacillus* M2.1A. Corresponding to the results obtained with *C. michiganensis*, *Lysinibacillus* CD3-6 did not suppress the bacterial plant pathogen (Appendix A).

The Gram-negative plant pathogens *Dickeya solani* and *Erwinia amylovora* (causative agent of fire blight disease at orchard trees) were also used as indicators for antagonistic activity. This ruled out the possibility that the Gram-negative bacteria were less inhibited by the *Brevibacillus* strains as the Gram-positive bacterium *C. michiganensis*. *Brevibacillus* sp. RS1.1, *Brevibacillus* sp. MS2.2, *B. porteri* HB1.4B, and *Brevibacillus* DP1.3A inhibited the growth of *Dickeya solani* and *Erwinia amylovora* after an incubation period of two or three days, but their inhibition zones were relatively small. *Brevibacillus* sp. HB2.2, *B. parabrevis* HB2.2, and HD1.4A did not apparently inhibit the growth of both Gram-negative pathogens, suggesting that Brevibacilli were more efficient against Gram-positive bacteria (Appendix A).

#### 3.3.2. Antifungal Activity

Antifungal activity was examined in vitro using four different *Fusarium* species (*F. oxysporum*, *F. culmorum*, *F. poae*, *F.graminearum*) known for causing fusarium wilt disease [46], and *Aspergillus niger* (‘black mold’). *Lysinibacillus* sp. CD3-6 was found to be inefficient against most of the *Fusarium* strains, but the *Brevibacillus* strains did suppress the growth of all phytopathogenic fungi and the oomycete *P. palmivora*, one of the most detrimental plant pathogens in Vietnam [47]. *B. porteri* HB1.4B, *Brevibacillus* sp. DP1.3A, *Brevibacillus* sp. RS1.1, and *Brevibacillus* sp. MS2.2 displayed strong inhibiting effects against *F. poae*, and to a minor degree against *F. culmorum*. *F. graminearum* was less sensitive, but was still inhibited by *Brevibacilllus* sp. HB2.2 and *Brevibacilllus* sp. RS1.1 (Appendix A).

#### 3.3.3. Nematicidal Activity

Root-knot nematodes, such as *Meloidogyne* spp., are one of the most important plant pathogens in tropical and temperate agriculture, and are responsible for significant harvest losses of main Vietnamese crops, such as coffee and black pepper [48]. In order to analyze the antagonistic activity of the *Brevibacillus* strains and *Lysinibacillus* CD3-6, we first tested their suppressing effect against the model nematode *Caenorhabditis elegans*. Fast and slow death rates were estimated in a bioassay under laboratory conditions. This ruled out the possibility that the *Brevibacillus* strains were much more efficient than *Lysinibacillus* sp. CD3-6. *Brevibacillus* sp. M2.1A displayed the highest killing effect against *C. elegans* (Figure 6A). In order to examine the suppressing effects against phytopathogenic nematodes more directly, we isolated a representative strain of *Meloidogyne* sp. directly from the galls of infested black pepper plant roots according to the hypochlorite procedure [49]. The suppressing effect exerted by the test bacteria on disease development was examined in a greenhouse experiment. Ten weeks after the transplanting of the tomato plantlets in soil, the formation of root knots was visually registered and used as a measure for calculating the disease index according to [27]. The *Brevibacillus* strains were found to be efficient in reducing the disease severity to around 50% and less compared to the untreated control. Again, *Brevibacillus* M2.1A performed the best, whilst *Lysinibacilllus* sp. CD3-6 was found to be less efficient than all tested *Brevibacillus* strains (Figure 6B).

Whilst reports about the biocontrol action of *Bacillus* ssp., such as *B. firmus* [50], *B. velezensis* former *B. amyloliquefaciens* [25], *B. cereus*, *B. thuringiensis*, and *B. subtilis* [51] are increasing, to the best of our knowledge we present here the first report on the nematicidal activity of Brevibacilli against root-knot nematodes.

#### 3.3.4. Plant Growth Promotion

We examined the effect of the *Brevibacillus* strains and *Lysinibacillus* CD3-6 in the *Arabidopsis thaliana* biotest system [28]. Several *Brevibacillus* strains enhanced the growth of the *Arabidopsis* seedlings in a considerable manner (Figure 7, Appendix A).

The highest increase in plant biomass was obtained for *B. porteri* HB1.2 (38.83%), *B. parabrevis* HD3.3A (34.93%), *B. porteri* HB1.1 (30%), and *B. parabrevis* HD1.4A (27.69%).

The above phenotypic experiment proved that plant-associated *Brevibacillus* strains isolated from Vietnamese crop plants are able to positively interact with *Arabidopsi*s plants, and to stimulate their growth in the range previously reported for the prototype of the Gram-positive plant-growth-promoting rhizobacteria, *Bacillus velezensis* FZB42 [52].

### 3.4. Genome Mining for Putative Natural Product Biosynthesis Gene Clusters

The identification of biocontrol actions directed against phytopathogenic bacteria, fungi and nematodes prompted us to investigate the biosynthetic potential of the isolates using either their draft or their full genome sequences. Antimicrobial compounds belong to structurally diverse groups of molecules, such as nonribosomal peptides (NRP) and polyketides (PK), and ribosomally synthesized and posttranslationally modified peptides [53]. The antiSMASH 6.0 version [29] was used for the prediction and annotation of the biosynthetic gene clusters (BGCs). The antiSMASH results were subsequently compared against the MIBiG database [54] in order to identify characterized and uncharacterized BGCs. In total, 151 BGCs in the 11 *Brevibacillus* isolates, and 6 BGCs in *Lysinibacillus* sp. CD3-6 were identified and separated into nine distinct classes (Appendix A). A total of 36 of them were not detected in the MIBiG database and might encode the biosynthesis of uncharacterized secondary metabolites. The edeine gene cluster, previously detected in *Brevibacillus brevis* Vm4 and X23 [55], but not listed in the MIBiG databank, was found to be widely distributed in the *Brevibacillus* spp. strains. A variant of another modular PK-NRP hybrid, Paenilipoheptin, recently described in *Paenibacillus polymyxa* E681 [56], was detected for the first time in the genus *Brevibacillus*.

#### 3.4.1. Gene Clusters Encoding Modular and Nonmodular Polyketides

Genes with more than 90% similarity to the gene cluster encoding the non-ribosomal synthesized polyketide macrobrevin [9] were detected in five of the Vietnam *Brevibacillus* strains (Appendix A). Generally, polyketides are synthesized by giant polyketide synthases in a non-ribosomal manner [57]. Similar to the gene cluster in *Brevibacillus* Leaf182 (BGC0001470), *Brevibacillus* sp. strains DP1.3A and HB1.3 contained the complete set of 15 modules, whilst strains MS2.2, HB2.2, and RS1.1, harboring only 12 modules, did probably encode truncated versions of the polyketide (Appendix A). Macrobrevin displays a unique polyketide structure, and was shown to be active against leaf colonizing *Bacillus* strains [9].

Gene clusters with weak similarity to the nonmodular Type III PKS gene cluster (BGC0001964) were detected in most *Brevibacilllus* strains, and also in *Lysinibacillus* sp. CD3-6 (Appendix A). Type III PKSs produce a large number of aromatic natural compounds in plants, fungi and bacteria. They catalyze the condensation of coenzyme activated starter units with (methyl)malonyl-CoA extender units by decarboxylative Claisen condensations [58]. A type III PKS chalcone synthase has been described in *Streptomyces griseus* [59] and other Gram-positive bacteria such as *B. subtilis* [60].

#### 3.4.2. Non-Ribosomal-Synthesized Antimicrobial Peptides (NRP)

Gene clusters exhibiting similarity with the genes encoding the well-known NRPs tyrocidine (BGC0000452), gramicidin (BGC0000367), and marthiapeptide (BGC0001469) were detected in different *Brevibacillus* strains isolated from Vietnam crop plants (Appendix A). NRPs are secondary metabolites which are synthesized through giant multi-modular peptide synthetases [61].

The three genes *tyc*A, *tyc*B, and *tyc*C encoding the cyclic ß-sheet decapeptide tyrocidine (C_66_H_87_N_13_O_13_) were detected in the 11 *Brevibacillus* strains representing the species *B. parabrevis*, *B.porteri* and four novel genomospecies (Appendix A). The gene clusters widely resembled the tyrocidine gene cluster from *Brevibacillus brevis* ATCC8185 [62]. Except for the adenylation domains present in modules 3 and 4, the other eight modules were found highly conserved in all *Brevibacillus* strains (Appendix A). This corresponds to the structures reported for tyrocidine A, B and C bearing residues Phe^3/4^, Phe^3^/Trp^4^, and Trp^3/4^, respectively. The membrane effective antibiotics kill Gram-positive bacteria such as *Bacillus subtilis* and *Staphylococcus aureus* [63].

Gene clusters, similar to the four gene cluster (*lgrA-lgrB-lgrC-lgrD*) described by [64] as being involved in non-ribosomal synthesis of the linear gramicidin peptides, were detected in *B. parabrevis* HD1.4A and HD3.3A, *Brevibacillus* sp. DP1.3A, HB2.2, RS1.1, and M2.1A (Appendix A). The complete gene clusters present in *B. brevis* (AJ566197), *B. parabrevis* HD3.3A (CP085874.1) HD1.4A (JABSUW010000001), and *Brevibacillus* sp. DP1.3A (CP085876) were characterized by a formylation (F) domain in the first module (*lgrA*) and a final reductase (TD) domain in the last module (*lgrD*). Two different types were distinguished (Figure 8, Appendix A). The gramicidin gene clusters in *B. parabrevis* HD1.4A and HD3.3A containing 16 modules were identical with the corresponding 74 kb *lgr* gene cluster in *B. brevis* ATCC 8185 (AJ566197). The product of the *lgr* gene cluster in ATCC 8185 is characterized as a linear gramicidin pentadecapeptide consisting of 15 hydrophobic amino acid residues with an alternating L- and D-configuration forming a β-helix-like structure. Module 16 is predicted to activate alanine (according to our antiSMASH prediction) or glycine (according to [64]), and subsequently to reduce alanine/glycine. Finally, *N*-formyl-pentadecapeptide-ethanolamine is released from the LgrD enzyme [64].

Surprisingly, we detected a novel variant of the gramicidin gene cluster in *Brevibacillus* sp. DP1.3A (GS A6-30) containing two additional modules in the 3′-region of the *lgrB* gene (Figure 8). Their adenylation domains were coding for Ala and D-Val, and the modules seem to be caused by a partial duplication in the 3′-region. We hypothesize that the 18-module gene cluster is coding for a linear heptadecapeptide (*N*-formyl-heptadecapeptide-ethanolamine) containing two additional amino acid residues, Ala and D-Val, in position 7 and 8. Other *Brevibacillus* strains harbored truncated forms of both gramicidin gene cluster variants. A truncated pentadecapeptide variant was detected in *Brevibacillus* sp. RS1.1, whilst gene clusters encoding truncated heptadecapeptides were detected in *Brevibacillus* sp. HB2.2, and MS2.2, belonging to genomospecies A6-33, and A6-34, respectively (Appendix A). Since the truncated gramicidin gene clusters were predicted from draft genomes (WGS), we cannot exclude the possibility that the deletions are due to incomplete sequences. The gramicidin gene cluster present in the MIBiG data bank, BGC0000367 (AP008955.1), was identified as a 13-module gene cluster bearing a 3′ deletion in the *lgrD* gene, probably encoding a truncated variant of the heptadecapeptide (Appendix A). We propose to replace BGC0000367 by the gramicidin gene clusters from *B. brevis* ATCC8185, encoding the complete gramicidin pentadecapeptide, and *Brevibacillus* sp. DP1.3A, encoding the putative gramicidin heptadecapeptide.

*Brevibacillus* sp. MS2.2. harbored a gene cluster with high similarity to marthiapeptide (BGC0001469), a gene cluster recently detected in *Brevibacillus* Leaf182 [9] (Appendix A). The polythiazole cyclopeptide marthiapeptide was first described in *Marinactinospora thermotolerans* and shown to inhibit Gram-positive bacteria and cancer cells [65].

#### 3.4.3. Gene Clusters Encoding PK-NRP Hybrids

Gene clusters, exhibiting high similarity to the edeine gene cluster in *Brevibacillus brevis* Vm4 (Appendix A), were detected in nearly all the *Brevibacillus* strains representing *B. porteri* (HB1.1, HB1.2, HB1.4B), and novel genomospecies such as A6-29 (MS2.2), A6-30 (DP1.3A), A6-33 (HB1.3, RS1.1), and A6-34 (M2.1A), but not in the *B. parabrevis* strains (Appendix A). The six genes *ede*P, *ede*N, *ede*L, *ede*K, *ede*J, and *ede*I encode NRPS and PKS-NRPS hybrids involved in the non-ribosomal synthesis of the atypical cationic edeine peptides (Appendix A). Edeines contain a ß-Tyr or a ß-Phe residue at the N-terminus and a spermidine-polyamine structure at the C-terminus, flanking five non-proteinogenic amino acids in the central part (Appendix A). Edeine is a broad-spectrum antimicrobial agent acting against bacteria and fungi. DNA synthesis is inhibited at low concentrations <15 μg/mL [66]. Despite the fact that edeines have been discovered as early as 1959 [67], corresponding gene clusters have not been deposited in the MIBiG data bank until now.

Interestingly, we found a gene cluster in *Brevibacillus* sp. DP1.3A, which strongly resembled the *phn* gene cluster in *Paenibacillus polymyxa* E681 (BGC0001728). The *phn* gene cluster is possibly involved in the non-ribosomal synthesis of the cyclic lipoheptapeptide paenilipoheptin [56]. Both gene clusters contain eight modules (Appendix A). The first module is involved in fatty acid synthesis, while the remaining modules, containing seven adenylation domains, were predicted to be responsible for the synthesis of a seven-member peptide chain containing D-ser- (1), D-phe- (5), tyr- (6), and glu-residues (7) (Figure 8B). The structure of paenilipoheptin synthesized by *P. polymyxa* was elucidated by MALDI-LIFT TOF/TOF MS as being C13-ß-NH2-FA—Ser (1)—Dab (2)—Trp (3)—Val (4)—Phe (5)—Tyr (6)—Glu (7) [56], which is compatible with the predicted structure of the gene cluster detected in *Brevibacillus* DP1.3A (Appendix A). A thioesterase domain at the 3′ end of the phnE gene was missing in both gene clusters, indicating that cyclization might be accomplished by a free-standing TE enzyme as proposed by Vater et al. [56].

#### 3.4.4. Gene Clusters Representing RiPPs, and Bacteriocins

In contrast to polyketides and peptides, which are synthesized independently from ribosomes, numerous secondary metabolites with antimicrobial activity such as RiPPs (ribosomally synthesized and posttranslationally modified peptides) and (unmodified) bacteriocins are synthesized by a ribosome-dependent mechanism. Several groups are distinguished [68]. Some of them, such as lassopeptides, LAPs, lanthipeptides, UviB peptides, and sactipeptides, were detected applying the antiSMASH and BAGEL4 [30] toolkits in the *Brevibacillus* isolates and/or the *Lysinibacillus* sp. CD3-6 strain (Figure 9).

Many RiPP biosynthetic proteins recognize and bind their cognate precursor peptide through a domain known as RiPP recognition element (RRE) [69]. The detection of RRE domains can be helpful in identifying gene clusters involved in the synthesis of novel classes of RiPPs [70], but does not necessarily identify a specific category of RiPPs. RRE containing domains with weak similarity to Pantocin-Microcin RRE (BGC 0000585) were found in nearly all *Brevibacillus* strains, but they were not associated with other genes involved in RiPP synthesis. In contrast, the gene for the lassopeptide RRE domain protein in *Lysinibacillus* sp. CD3-6 was associated with the genes encoding the lassopeptide class ii core (leader) peptide with a putative macrolactam sequence, and the lasso peptide biosynthesis B2 protein (Appendix A).

Gene clusters encoding linear azol(in)e-containing peptides (LAPs) were detected in all 11 *Brevibacillus* isolates, and *Lysinibacillus* CD3-6 (Appendix A). Most of them were identified as being members of the TOMM class (thiazole/oxazole-modified microcins) characterized by a gene cluster consisting of a cyclodehydratase gene and associated genes encoding dehydrogenase and a maturation protein. A gene encoding a TOMM precursor leader peptide was detected upstream of these three genes (Appendix A). Typically, the TOMM precursor leader peptides were characterized by a homologous leader region and then a region enriched with Cys residues, a feature of the hetero-cycloanthracin/sonorensin family [68]. This type of thiopeptide encoding gene was detected in representatives of the *Brevibacillus* genomospecies 25, 29, 31, and 33 (Appendix A). Different sequences encoding TOMM leader peptides enriched with Val-Ala or Ser were detected in *Brevibacillus parabrevis* (GS 25), *Brevibacillus* sp. DP1.3A (GS 30), and *Brevibacillus* sp. M2.1A (GS34) (Appendix A). The LAP gene cluster in *Lysinibacillus* CD3-6 encoded a SagB/ThcOx family dehydrogenase, and a YcaO-like family protein, but did not possess genes encoding maturase and precursor peptide proteins.

Ripps similar to Linocin M18 were found in all *Brevibacillus* GS (Appendix A). The Linocin_M18 bacteriocin, first isolated from *Brevibacterium linens* M18 by Valdez-Stauber and Scherer [71], inhibits Gram-positive bacteria. These widely distributed proteins, referred to as encapsulins, form nano-compartments within the bacterium which contain ferritin-like proteins or peroxidaseenzymes. Lanthipeptides are defined by the presence of ß-thioether cross-links, which are generated by the posttranslational modification of Ser/Thr and Cys residues [72]. Best studied are class I and class II lanthipeptides, which are modified by different dehydratases (LanB or LanM) and cyclases. Gene clusters encoding class I lanthipeptides were detected in two representatives of *Brevibacillus* GS A6-33 (HB1.3, RS1.1) and GS A6-34 (M2.1A) (Appendix A). Genes involved in synthesis and posttranslational modification of class II lanthipeptides were only detected in *Brevibacillus* DP1.3A, a representative of GS A6-30 (Appendix A). Gene clusters for biosynthesis of class III lanthipeptides were detected in *B. porteri* strains HB1.1 and HB1.2, and in *Lysinibacillus* CD3-6 (Appendix A).

A representative of UV inducible peptides (UviB) was detected in *Brevibacillus parabrevis* using BAGEL4 supported genome mining (Appendix A). The Phage_holin_BhlA family is a family of holin-like proteins from both bacteriophages and bacterial chromosomes. BhlA, a putative holin-like protein of *Bacillus licheniformis* AnBa9 [73], and *Bacillus pumilus* WAPB4 [74], showed antibacterial activity against several Gram-positive bacteria.

#### 3.4.5. Gene Cluster Involved in Synthesis of Siderophores and Other BGCs

The *asbABCDEF* gene cluster (BGC0000942) from *Bacillus anthracis*, *the causative agent of anthrax*, is responsible for the biosynthesis of petrobactin, a catecholate siderophore that functions in both iron acquisition and virulence [75]. It has been argued that the iron-siderophore petrobactin contributes to *B. anthracis* pathogenesis, which requires maintaining sufficient iron concentration during growth in the host tissue [76]. However, it has been documented that petrobactin synthesis is also common in non-pathogenic members of the *B. cereus senso latu* group [77]. Except for two representatives of *B. parabrevis* (HD1.4A, HD3.3A), we detected gene clusters with apparent similarity to BGC0000942 in representatives of *B. porteri* (HB1.1, HB1.2, HB1.4B) and the novel GS A6-29 (MS2.2), A6-30 (DP1.3A), and A6-33 (HB2.2, RS1.1, HB1.3) (Appendix A). By contrast, the gene clusters encoding non-ribosomal synthesis of the second siderophore bacillibactin, common in the *B. subtilis*, and *B. cereus* group, were not detected in the investigated *Brevibacillus* strains.

Other BGCs such as gene clusters involved in synthesis of terpenes, and the cyclic lactone autoinducer with a putative role in quorum sensing (AgrB precursorpeptides) were detected in most *Brevibacillus* isolates. *Lysinibacillus* sp. CD3-6 also harbored terpene encoding genes, and a gene cluster possibly involved in the synthesis of a betalactone containing protease inhibitor (Appendix A).

#### 3.4.6. Uncharacterized NRP and PK-NRP Hybrid Gene Clusters in *Brevibacilli*

In addition to the characterized BGCs mentioned above, we detected 35 hitherto unknown NRPs and NRP-PK hybrid scaffolds in the *Brevibacillus* genomes, and one partial NRPS gene cluster containing a tyrosine module in *Lysinibacillus* CD3-6 (Appendix A).

A putative PK-NRP hybrid consisting of four modules (PK-orn-x-phe) occurred in all nine *Brevibacillus* strains belonging to the *Brevibacillus* A6-branch. In contrast, a three-module hybrid (asn-gly-pk) occurred only in *Brevibacillus* M2.2 (Appendix A).

Giant PK-NRP hybrids consisting of either 16, 20, or 25 modules (Appendix A) were detected in several *Brevibacillus* strains. The 16 M scaffold consisted of 14 PK modules (mal, ccmal, ohmal), and two modules involved in the non-ribosomal synthesis of amino acids (M6:X, M16: Ser). The three *B. porteri* strains, and *Brevibacillus* HB1.3, RS1.1, HB2.2, and M2.1.A harbored all 16 modules of the hybrid, whilst *Brevibacillus* DP1.3A harbored a deleted gene cluster, in which the modules 9–16 were missing. The largest gene cluster harboring 25 modules was detected in *B. parabrevis* HD3.3A. Some 14 modules were responsible for the non-ribosomal synthesis of amino acids, whilst the 11 pk-modules were involved in the synthesis of mal, ohmal, and ccmal. A deleted form of this scaffold harboring only 20 modules was detected in *B. parabrevis* HD1.4A.

Decapeptide scaffolds were detected adjacent to the *B. parabrevis* giant M25/20 PK-NRP hybrid scaffolds (Appendix A). Experimental proof is needed to corroborate their stand-alone state. Alternatively, it is also possible that they are directly connected with the giant hybrids.

Slightly modified NRP scaffolds consisting of either six or seven modules, and always starting with Glu-Ser at their N-terminus, were present in nearly all representatives of the A6 branch (Appendix A). The six-module variant (glu-ser-ile-x-phe-D-orn) occurred in the genomes of *Brevibacillus* M2.2, and HB2.2. One of the two seven-module scaffolds contained the predicted sequence Glu-Ser-Val-Val-X-Phe-D-Orn, and occurred in *Brevibacillus* sp. DP1.3A, HB1.1, HB1.2, HB1.3. *Brevibacillus* sp. M2.1A harbored a second variant characterized by the predicted sequence Glu-Ser-Val-X-X-Phe-D-Orn. Preliminary experiments did not reveal hexa- or heptapeptides in the *Brevibacillus* strains, but pentapeptides with partial corresponding sequences were detected. At present, we cannot exclude the possibility that the scaffolds are responsible for the synthesis of the pentapeptides, but one or two of the modules might be overread or not expressed.

An interesting NRP gene cluster encoding for three cysteine residues was detected in *B. parabrevis* HD1.4A, HD3.3A, *Brevibacillus* DP1.3A, and *B. porteri* HB1.1, HB1.2, HB1.4B (Appendix A). Due to the presence of predicted methyltransferase domains, it can be assumed that the cysteine residues are methylated (Appendix A). The cryptic *nrs* gene cluster of the model biocontrol bacterium *Bacillus velezensis* FZB42 also harbors three modules encoding for cysteine. Recently, trithiazole was identified as the product of the *nrs* gene cluster, and NrsB as the oxidizing enzyme of the thiazoline precursor [78]. However, a more careful comparison of the domain structure of both gene clusters revealed significant differences between both BGCs, excluding the possibility that the final product can be a polythiazole (Figure 10). Although no o Ox domain was detected, two genes involved in dihydroxybenzoate (dhb) synthesis were (Figure 10A). The 2,3-dhb-AMP ligase and an oxido-reductase, predicted to be involved in dhb synthesis, possess counterparts in the *dhb* gene cluster of FZB42 known to be responsible for the non-ribosomal synthesis of the siderophore bacillibactin (BGC0000616.1, Figure 10C). For this reason, we assume that the unknown gene cluster might be involved in the biosynthesis of a novel siderophore a with function in iron acquisition.

## 4. Conclusions

Recently a total of 59 endospore-forming Gram-positive bacteria were isolated from healthy crop plants within fields in Vietnam that were infested with plant-pathogenic nematodes and fungi. According to their draft genome sequences, the majority of the strains were classified as being members of the *B. subtilis* group, mainly *B. velezensis*, and the *B.cereus* group. The remaining 12 strains were provisionally classified as *Brevibacillus* sp. and *Lysinibacillus* sp., respectively [1]. In this study we have focused on the members of the *Brevibacillus* and *Lysinibacillus* taxon, and a special procedure for enrichment of *Brevibacillus* strains was developed. The sequencing of the whole genomes of *Lysinibacillus* sp. CD3-6 and of three *Brevibacillus* strains allowed for a detailed genome analysis of the selected strains, and the identification of several extrachromosomal elements. The novel isolate *Brevibacillus* sp. DP1.3A, for example, harbored three different plasmids, representing either the low-copy type, whose segregation was directed by ParM, or the high copy type, replicating according to the rolling circle model. The high plasticity of the genomes was also indicated by the presence of numerous genomic islands within the *Lysinibacillus* and *Brevibacillus* chromosomes, and the high number of genes present in the pan-genomes.

Furthermore, we have elucidated the taxonomic position of the eleven *Brevibacillus* strains, and of the *Lysinibacillus* strain on the base of their complete or draft genome sequences. Five of the *Brevibacillus* strains were classified as being members of the known species, *B. parabrevis* (2), and *B. porteri* (3). The other six *Brevibacillus* strains, and *Lysinibacillus* CD3-6 were, according to their ANI and dDDH values, classified as being members of five novel genomospecies.

The main outcome of this work was characterizing the *Brevibacillus* isolates as potent antagonists of important plant pathogens. *Brevibacillus* strains were found to be efficient in directly inhibiting the growth of bacterial phytopathogens, such as *Clavibacter michiganensis*, *Xanthomonas campestris*, *Erwinia amylovora*, *Dickeya solani*, and fungal pathogens, such as *Fusarium oxysporum*, and other representatives of the *Fusarium* genus. The direct antagonistic action of the *Brevibacillus* strains against the phytopathogenic oomycete *Phytophthora palmivora* was also observed. Brevibacilli displayed high nematicidal activity, and significantly suppressed the formation of root-knots in tomato plants infested with *Meloidogyne* sp. in greenhouse experiments under controlled conditions.

The *Brevibacillus* genomes harbored a rich arsenal of known and hitherto uncharacterized BGCs predicted to synthesize ribosomally and non-ribosomally diverse classes of secondary metabolites with potential antagonistic action against plant pathogens. A total of 151 BGCs including 35 uncharacterized BGCs were identified by genome mining in the *Brevibacillus* genomes, whilst *Lysinibacillus* sp. CD3-6 harbored only six BGCs. The occurrence of important BGCs in the representatives of different genomospecies is shown in Figure 11. In several cases, it seems that the distribution of BGCs in Brevibacilli is dependent on their taxonomical position. Gene clusters for the synthesis of edeine and petrobactin, for example, were not detected in *B. parabrevis*, but were present in the species cluster A29, A30, A31, A33, and A34. The gene clusters devoted to tyrocidine synthesis were identified in all isolates. In contrast, gene clusters involved in the synthesis of marthiapeptide and paenilipoheptin were only detected in single strains representing the species cluster A29 and A30, respectively. The paenilipoheptin gene cluster exhibited striking similarity to a corresponding gene cluster in *Paenibacillus polymyxa* [56] Our results are in line with the previous finding that *Brevibacillus* sp. Leaf182 is the most potent antagonist within the *Arabidopsis* phyllosphere-microbiome consisting of 224 strains [9]. The BGC responsible for non-ribosomal synthesis of the antibacterial polyketide macrobrevin, originally described in Leaf182, was detected in three novel genomospecies (A29, A30, and A33). Its presence on the DP1.3A genomic islands suggested that the cluster could be transferred by horizontal gene transfer. Two types of gramicidin gene clusters were detected. One of them which harbored 18 modules has been not described previously. In summary, our genome mining results indicated that Brevibacilli are a treasure box of widely unexplored AMPs and other secondary metabolites. Biocontrol agents developed from plant-associated *Brevibacillus* strains should extend our present arsenal of bio-based plant protection agents that are necessary for a more sustainable agriculture.

## Figures and Tables

**Figure 1 microorganisms-11-00168-f001:**
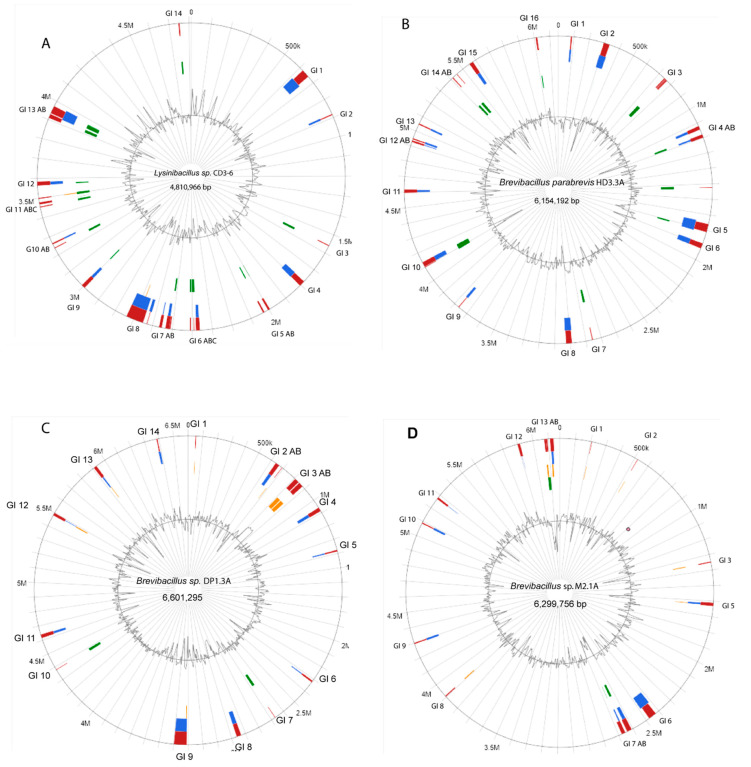
Circular plots of the chromosomes of *Lysinibacillus* sp. CD3-6 (**A**), *B. parabrevis* HD3.3A (**B**), *Brevibacillus* sp. DP1.3A (**C**), and *Brevibacillus* sp. M2.1A (**D**). Predicted GIs are shown as blocks colored according to the prediction method; Island Pick (green), Island-Path-DIMOB (blue), SIGI-HMM (orange), as well as the integrated results (dark red). The grey line within the inner circle shows deviations of the average GC-content.

**Figure 2 microorganisms-11-00168-f002:**
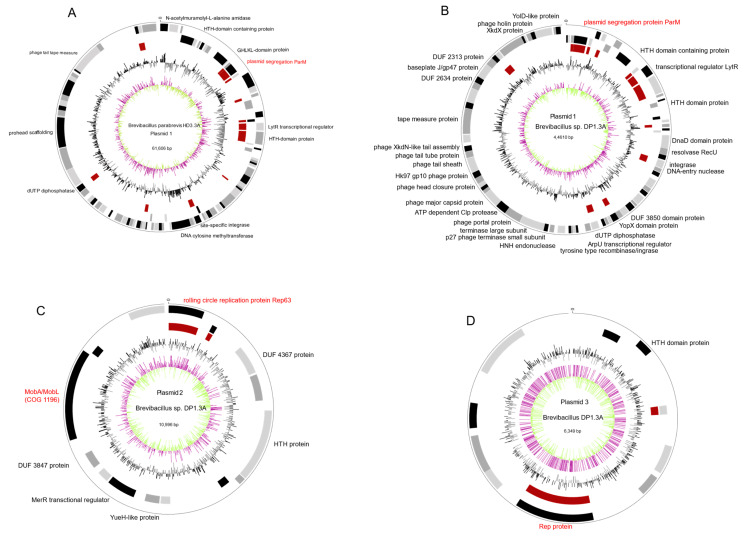
Circular plot of *Brevibacillus* plasmids generated with Biocircos. From outer to inner circle: Genes (CDS) on + (1)/− strand (2); core genome, brown (3); GC-content (1000 bp window) above mean: black, below mean: grey (4); GC Skew [(G-C)/(+C9] (1000 bp window), above mean: purple, below mean: light green (5). (**A**): Plasmid 1 (*B. parabrevis* HD3.3A) computed against plasmid 1 (*Brevibacillus* sp. DP1.3A). (**B**): Plasmid 1 (*Brevibacillus* sp. DP1.3A) computed against plasmid 1 (*B. parabrevis* HD3.3A). (**C**): Plasmid 2 (*Brevibacillus* sp. DP1.3A) computed against plasmid 3 (*Brevibacillus* sp. DP1.3A). (**D**): Plasmid 3 (*Brevibacillus* sp. DP1.3A) computed against plasmid 2 (*Brevibacillus* sp. DP1.3A).

**Figure 3 microorganisms-11-00168-f003:**
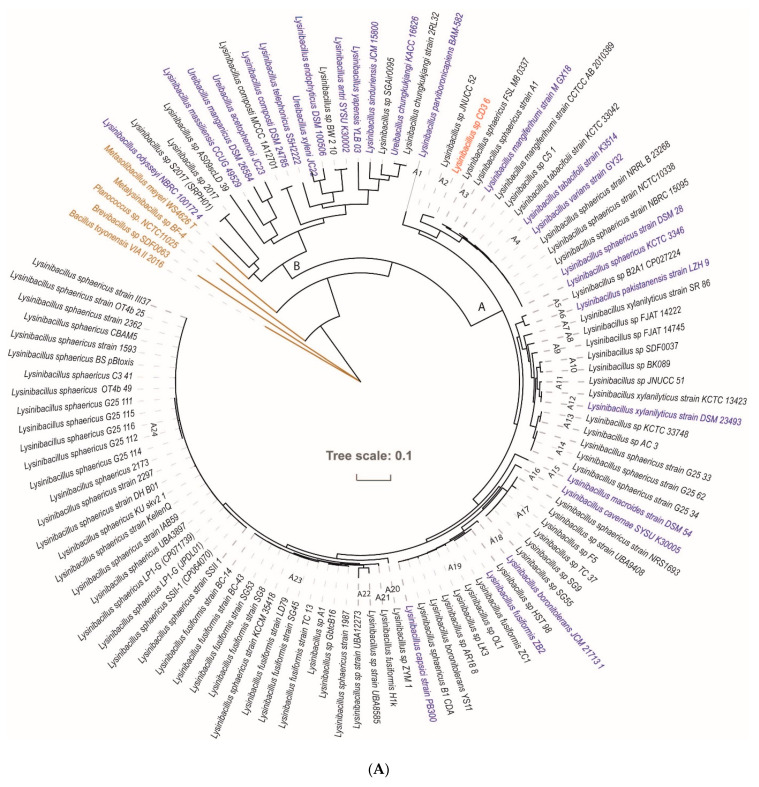
(**A**). Approximately-maximum-likelihood phylogenetic tree for 113 *Lysinibacillus* genomes, calculated by EDGAR using the Fast Tree software (http://www.microbesonline.org/fasttree/, accessed on 3 December 2022) and drawn by iTOL. Type strains are labelled in blue letters. Strains, previously misidentified as *Lysinibacillus* and now reclassified as representatives of other species, are labelled in brown. The tree harbored two main branches. Branch A contains the species related to *Lysinibacillus sphaericus*, whilst branch B contains the species which are more remote from *L. sphaericus*. The tree was built out of a core of 156 genes per genome, for a total of 17,628. The core has 58,525 AA-residues/bp per genome, for a total of 6,613,325. (**B**). *Lysinibacillus* tree inferred with FastMe 2.1.6.1 from GBDP distances calculated from whole genome sequences using the Type (Strain) Genome Server TYGS (https://tygs.dsmz.de, accessed on 3 December 2022). The tree consisted of 26 species and 28 subspecies clusters. Analysis was performed using both Maximum Likelihood and Maximum Parsimony. The numbers above the branches are GBDP pseudo-bootstrap support values >60% from replications, with an average branch support of 66.0%. Genomospecies according to (**A**) are indicated. The first two colored columns to the right of each name refer to the genome-based species and subspecies clusters, respectively, as determined by dDDH cut-off of 70 and 79%, respectively. The GTDB species are indicated at the right. The clustering yielded 16 species clusters and strain CD3-6 (labelled in red) was assigned together with JNUCC-52 as novel genomospecies (A2, *Lysinibacillus sp002340205*). Type strains are labelled in blue letters. The tree was rooted at the midpoint.

**Figure 4 microorganisms-11-00168-f004:**
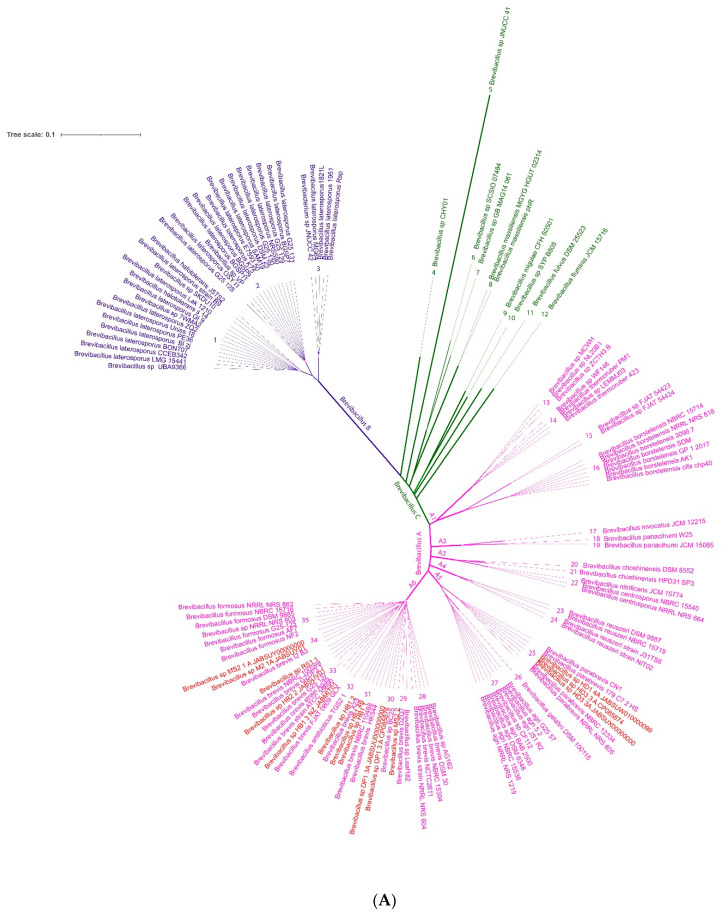
(**A**). Approximately-maximum-likelihood phylogenetic tree for 134 *Brevibacillus* genomes using the FastTree software accessible within the EDGAR package 3.0 [16]). Brevibacilli investigated in this study are labelled in red letters. The tree was built out of a core of 495 genes per genome, for 66,330 in total. The core has 167,683 AA-residues/ bp per genome, 15,461,980 in total. To construct a phylogenetic tree for a project, the core genes of these genomes are computed. In a following step, alignments of each core gene set are generated using MUSCLE, and the alignments are concatenated to one huge alignment. (**B**). *Brevibacillus* tree inferred with FastMe 2.1.6.1 [44] from GBDP distances calculated from whole genome sequences using the Type (Strain) Genome Server TYGS (https://tygs.dsmz.de, accessed on 3 December 2022). Analysis was performed using both Maximum Likelihood and Maximum Parsimony, with 14 type strains (labelled in blue letters) and 48 additional genome sequences including the *Brevibacillus* strains isolated from Vietnamese crop plants (labelled by red letters). The numbers above branches are GBDP pseudo-bootstrap support values >60% from replications, with an average branch support of 85.1%. The first column to the right of each name refers to the genome-based species according to the nomenclature given in (**A**). Species and Subspecies cluster (columns 2 and 3) are characterized by dDDH cut-off values of 70 and 79%, and ANI values of 96 and 98%, respectively. A total of 15 GTDB species clusters could be distinguished. The tree was rooted at the midpoint.

**Figure 5 microorganisms-11-00168-f005:**
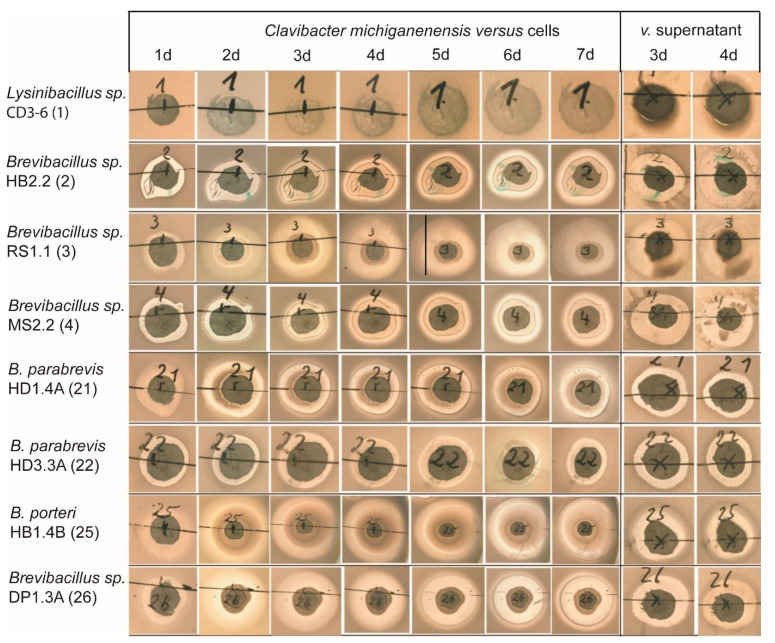
Inhibition of *Clavibacter michiganensis* by *Lysinibacillus* and *Brevibacillus* cells and supernatants indicated by clearance zones around the test bacteria.

**Figure 6 microorganisms-11-00168-f006:**
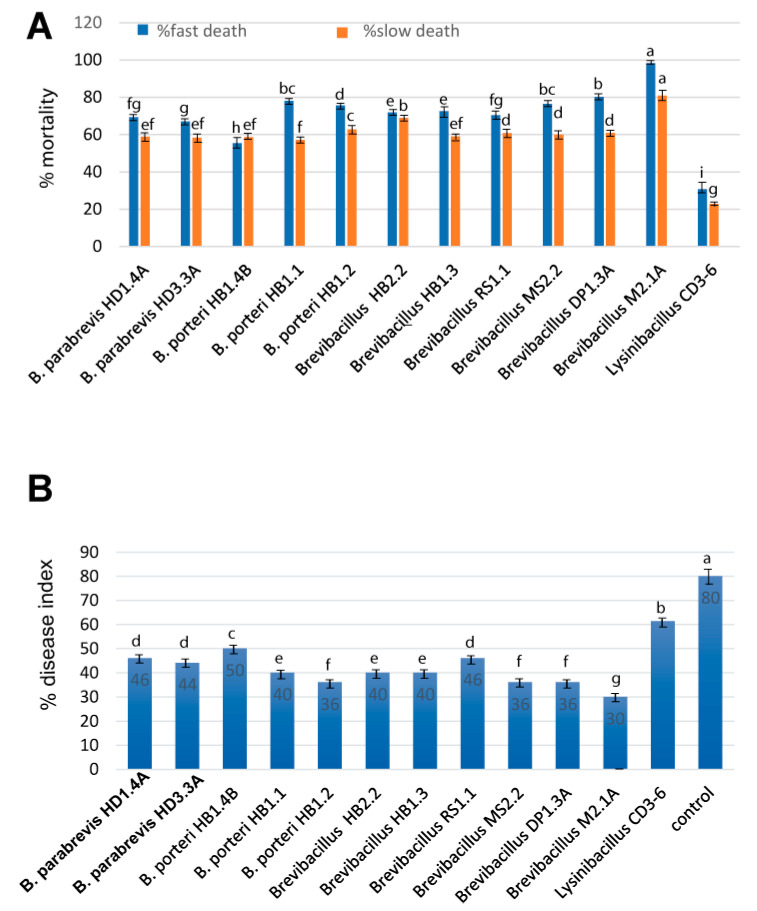
Nematicidal activity of *Brevibacillus* strains and *Lysinibacillus* CD3-6. (**A**): Bioassay with *Caenorhabditis elegans*. Slow killing activity was determined on NGM plates, and fast killing activity in liquid medium. (**B**): Tomato plants were infested with the root-knot nematode *Meloidogyne* sp. and the root-knot-forming rate in the presence and absence of the bacterial strains was estimated after 10 weeks growth of the tomato plants under controlled conditions in a greenhouse. All experiments were carried out with three independent repetitions and a completely randomized design. Different letters at each treatment indicate the significance between inoculated and uninoculated conditions at the *p* ≤ 0.05 level after the *t*-test.

**Figure 7 microorganisms-11-00168-f007:**
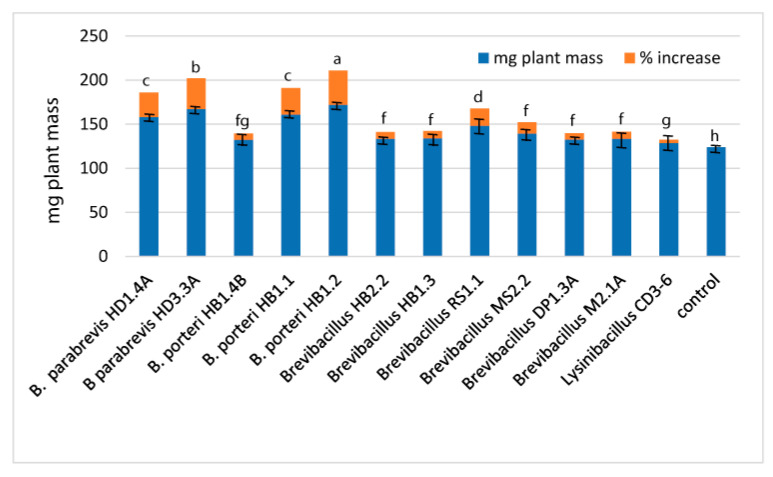
Growth promoting effects of *Brevibacillus* strains and *Lysinibacillus* CD3-6 on *Arabidopsis thaliana* seedlings. The % increase compared to the untreated control is indicated on top of the columns. Each treatment value is presented as the means of three replications (n = 3) with SE. Different letters at each treatment indicate the significance between inoculated and uninoculated conditions at the *p* ≤ 0.05 level after the *t*-test.

**Figure 8 microorganisms-11-00168-f008:**
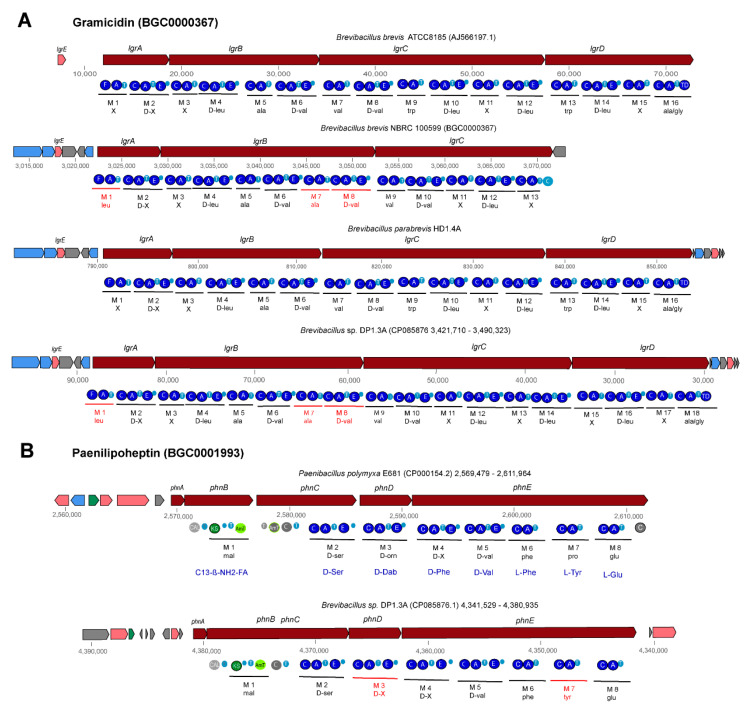
Gene clusters involved in non-ribosomal synthesis of peptides and lipopeptides detected in *Brevibacillus* genomes by antiSMASH. (**A**): Two types of gramicidin gene clusters were detected in the *Brevibacillus* isolates: a 16-module gene cluster in *B. parabrevis* HD1.4A encoding a putative *N*-formyl-pentadecapeptide-ethanolamine, and an 18-module gene cluster in *Brevibacillus* sp. DP1.3A encoding a putative *N*-formyl-heptadecapeptide-ethanolamine. (**B**): The gene cluster located in region 15 (4,380,935–4,341,529) of *Brevibacillus* sp. DP1.3A resembled BGC0001728, and is possibly involved in the non-ribosomal synthesis of paenilipoheptin in *Paenibacillus polymyxa* E681. Different modules are labelled in red. NRPS/PKS domains are indicated by filled circles. A: adenylation, C: condensation, CAL: co-enzyme A ligase, E: epimerization domain, KS: ketosynthase domain, T: thiolase.

**Figure 9 microorganisms-11-00168-f009:**
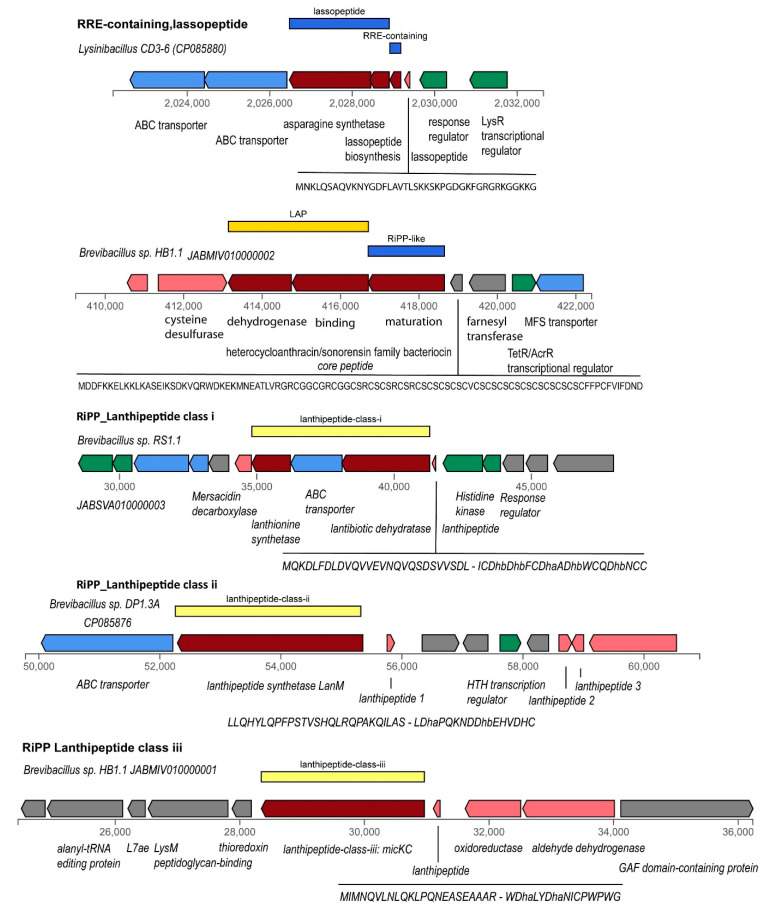
Different classes of RiPPs occurring in *Lysinibacillus* CD3-6 and in the plant-associated *Brevibacillus* strains.

**Figure 10 microorganisms-11-00168-f010:**
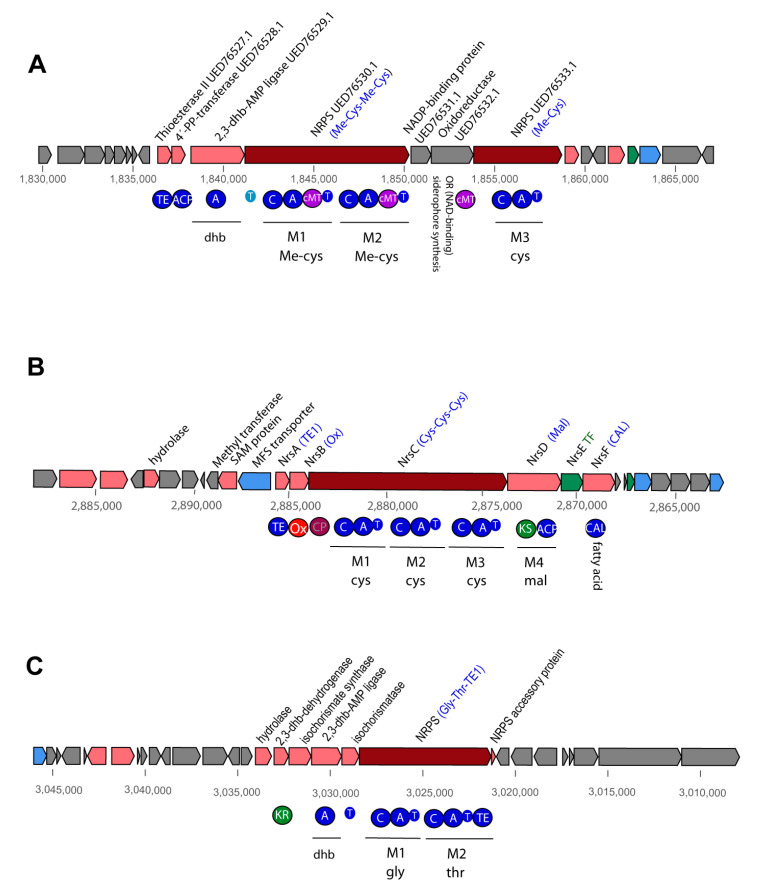
The uncharacterized NRPS gene cluster in *Brevibacillus* sp. DP1.3A (**A**) compared with the Bacillothiazol synthesizing *nrs* gene cluster (**B**) and the siderophore bacillibactin *dhb* gene cluster in *Bacillus velezensis* FZB42 (**C**). NRPS/PKS domains are indicated by filled circles. A: adenylation, ACS: 4′-phosphopantetheinyl transferase, C: condensation, CAL: co-enzyme A ligase, cMT: carbon methyltransferase, KR: ketoreductase, KS: ketosynthase, T: thiolase, TE: thioesterase.

**Figure 11 microorganisms-11-00168-f011:**
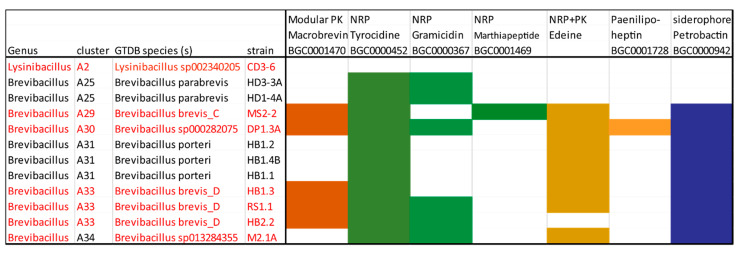
Occurrence of important BGCs in the different genomospecies investigated in this study. Novel genomospecies without type strain are indicated by red letters.

## Data Availability

Accession numbers of DNA sequences are given in Section 2.6.

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
