# Peer review of "Novel Plant-Associated *Brevibacillus* and *Lysinibacillus* Genomospecies Harbor a Rich Biosynthetic Potential of Antimicrobial Compounds"

_microorganisms, 2023, doi:10.3390/microorganisms11010168_

Round 1
Reviewer 1 Report
This paper describes bacterial strains belonging to the genera Brevibacillus and Lysinibacillus, which were isolated from agricultural plants in Vietnam. These strains show an antagonistic activity against several phytopathogens and promote plant growth. They may be used in agriculture to control plant diseases.
This study was three-folded: whole genome sequencing and taxonomy of the isolates; experimental trials on suppression of the phytopathogens and plant promotion; identification of genes encoding synthesis of several antibiotics and bioactive compounds. The study was comprehensive, the methodology was adequate.
One drawback is that these three parts of the study are not properly linked to each other. This problem can be solved if the conclusions where instead of repeating what was said already in the results, the authors could give a summary on whether the beneficial activities were related to the taxonomic position of the strains or to the presence of the specific genes/operons in their genomes. Additionally, instead of Fig. 3, which can be moved to the supplementary materials, it’d be better to have here a table with numbers of antagonistic activity of the strains in the form of the diameters of the inhibition zones +/- standard deviations (histograms like in Figure 7 can be an alternative). At the end of the section 3 or in the conclusion, a table should be added to demonstrate the distribution of genes encoding bioactive secondary metabolites in the genomes of bacteria belonging to different genospecies to illustrate whether this distribution follows the taxonomy of the strains or was it random and strain-specific? This table may include not all the genes discussed in the section, but at least several the most important and indicative in terms of the expected bioactivity.
The use of the terms is not always consistent. Gram-positive and Gram-negative must start with a capital letter that was not the case in the lines 456, 459, 475. Use everywhere either Brevibacillus Leaf182 or Leaf 182.
Line 375: “The method based on a BLASTN comparison of the genome sequences”. Should it be “method was based…”?
Line 128: The title of the section 2.3 merged with the previous paragraph.
Author Response
Author's Reply to the Review Report (Reviewer 1)
This paper describes bacterial strains belonging to the genera Brevibacillus and Lysinibacillus, which were isolated from agricultural plants in Vietnam. These strains show an antagonistic activity against several phytopathogens and promote plant growth. They may be used in agriculture to control plant diseases.
This study was three-folded: whole genome sequencing and taxonomy of the isolates; experimental trials on suppression of the phytopathogens and plant promotion; identification of genes encoding synthesis of several antibiotics and bioactive compounds. The study was comprehensive, the methodology was adequate.
One drawback is that these three parts of the study are not properly linked to each other. This problem can be solved if the conclusions where instead of repeating what was said already in the results, the authors could give a summary on whether the beneficial activities were related to the taxonomic position of the strains or to the presence of the specific genes/operons in their genomes. Additionally, instead of Fig. 3, which can be moved to the supplementary materials, it’d be better to have here a table with numbers of antagonistic activity of the strains in the form of the diameters of the inhibition zones +/- standard deviations (histograms like in Figure 7 can be an alternative). At the end of the section 3 or in the conclusion, a table should be added to demonstrate the distribution of genes encoding bioactive secondary metabolites in the genomes of bacteria belonging to different genospecies to illustrate whether this distribution follows the taxonomy of the strains or was it random and strain-specific? This table may include not all the genes discussed in the section, but at least several the most important and indicative in terms of the expected bioactivity.
Response: I followed the proposal of Reviewer 1 and have included an additional figure (figure 11), which demonstrates that the distribution of BGCs in Brevibacilli is in several cases dependent on their taxonomical position. The text in the conclusion part was improved in order to show that taxonomic position and occurrence of the gene clusters involved in synthesis of secondary metabolites are connected with each other.
The use of the terms is not always consistent. Gram-positive and Gram-negative must start with a capital letter that was not the case in the lines 456, 459, 475. Use everywhere either Brevibacillus Leaf182 or Leaf 182.
Response: Corrections were performed. Gram-positive and Gram-negative bacteria and Leaf182 are now the correct designations.
Line 375: “The method based on a BLASTN comparison of the genome sequences”. Should it be “method was based…”?
Response Line 350/375: The sentence was changed to:” The method was based on a BLASTN comparison of the genome sequences [43].”
Line 128: The title of the section 2.3 merged with the previous paragraph.
Response: Paragraphs 2.2 and 2.3 are now merged according to the proposal given by Reviewer 1.
Reviewer 2 Report
This works extensively describes the genomic features of Brevibacillus and Lysinibacillus strains associated with plants, especially the presence of plasmids and secondary metabolites related to the observed antagonistic effects. Antagonistic assays were performed against bacteria, fungi, oomicetes and nematodes. Genome metrics ANI and dDDH suggested the existence of new genomospecies within these genera. Experiments seem very well performed, described and results are well detailed. This work contributes with the description of the biotechnological potentials of two genera still not widely explored. My main concerns are related to the taxonomic analysis and presentation of supplementary material, which is useful but confusing.
TAXONOMY
I understood that both ANI and dDDH were used to classify strains. However, dDDH values were not shown.
Besides that, type strains should always be highlighted in figures and tables.
Please see more details below:
line 345- Were these 24 clusters recognized once they met both ANI and dDDH criteria? Please provide dDDH values, at least for these 24 clusters.
Please explain the criteria used to recognize subspecies and provide results. I believe there are 28 suggested subspecies in your figure, please revise.
I understood that there were more than two genomes that could not be assigned to known species. Please explain the legend.
Please indicate all type strains, DSM28T is not indicated in your figure.
line 375- 376- Please revise this sentence, it is not understandable.
Fig 4 and 5. Please indicate the type strains. Fig 5. Explain the meaning of using species names with blue and black labels.
Table S7. ANI and dDDH values should be provided. Explain what are the numbers provided in the "remarks" column.
SUPPL. MATERIAL & FIGS
Since there are many results, figures and tables should be carefully prepared to clearly show the results.
Please revise figures and especilly the tables since they seemed in an unfinished format. Suppl. material should be prepared with similar care as main material.
Different colors, fonts and highlights were used without explaining the meaning. Some info are duplicated. These makes it more confusing.
Suppl. figs
Fig S1-S4. What does "high" similarity mean? Provide an objective criteria.
It is not possible to recognize RNA and MIsc features.
It is not necessary to show genome names twice in the figure. The legend of the figure could be used for that.
Suppl. tables
Please explain the meaning of the colors applied in tables S1-S4.
Table S6. Some genomes present ANI values of 100 and there is no other genome in the same cluster. What does this value mean? This table is not very clear.
Suppl. Fig S12- Please provide the meaning of the abbreviated modules in the legend. Please explain why are some modules with faded colors and others with an "X". Were these modules completely sequenced or were they in the edge og the contig?
line 711- Suppl. Fig. S21?
Fig 11. The information shown in blue is duplicated. Likewise, figures A, B and C have legends in the figure that should be shown in the legend only. Please check all your figures accordingly.
OTHERS
line 489- Phytophthora is an oomytece, not a fungus.
Author Response
Author's Reply to the Review Report (Reviewer 2)
This works extensively describes the genomic features of Brevibacillus and Lysinibacillus strains associated with plants, especially the presence of plasmids and secondary metabolites related to the observed antagonistic effects. Antagonistic assays were performed against bacteria, fungi, oomicetes and nematodes. Genome metrics ANI and dDDH suggested the existence of new genomospecies within these genera. Experiments seem very well performed, described and results are well detailed. This work contributes with the description of the biotechnological potentials of two genera still not widely explored. My main concerns are related to the taxonomic analysis and presentation of supplementary material, which is useful but confusing.
TAXONOMY
I understood that both ANI and dDDH were used to classify strains. However, dDDH values were not shown.
Response: The dDDH values were used in the analysis provided by the TYGS server and are now presented together with the ANI values in Suppl. Table S6 and S7.
Besides that, type strains should always be highlighted in figures and tables.
Response: The TYGS server was used as main tool for taxonomic classification. The server uses type strains for taxonomic classification. User strains involved in analysis were labelled throughout with blue letters (Fig. 3A, Fig. 3B, Fig. 4A and 4B).
Please see more details below:
line 345- Were these 24 clusters recognized once they met both ANI and dDDH criteria? Please provide dDDH values, at least for these 24 clusters.
Response: The dDDH values used for classification of the 24 branch A Lysinibacillus clusters are presented together with their corresponding ANI values in Suppl. Table S6. The same was done in Suppl. Table S7 for the Brevibacillus clusters A13 – A35.
Please explain the criteria used to recognize subspecies and provide results. I believe there are 28 suggested subspecies in your figure, please revise.
Response: The dDDH values used for species delineation were ≤79% (see Fig. 4B), The results were presented in Fig. 3B (Lysinibacillus) and 4B (Brevibacillus). The number of subspecies was corrected in Fig. 3B to 28, since strain Lysinibacillus sp. UBA 8585, which represents clusterA22, failed the quality check in the GTDB data bank (Suppl. Fig. S8).
I understood that there were more than two genomes that could not be assigned to known species. Please explain the legend.
Response: In case of Lysinibacillus the novel isolate CD3-6 represents a novel genomospecies A2 (Fig. 3A and 3B). In case of Brevibacillus four clusters, namely A34 (Brevibacillus sp. M2.1A), A33 (isolates RS1.1, HB2.2, HB1.3), A30 (DP1.3A), and A29 (MS2.2) represent novel genomospecies. These results are summarized at the end of section 3.2.
Please indicate all type strains, DSM28T is not indicated in your figure.
Response: all type strains including DSM28 were indicated in the corresponding figures 3A and B, and 4A and B using blue letters.
line 375- 376- Please revise this sentence, it is not understandable.
Fig 4 and 5. Please indicate the type strains. Fig 5. Explain the meaning of using species names with blue and black labels.
Response: Blue means type strains, black means other strains used for constructing the phylogenetic tree. They were taken from the NCBI data bank. Red means the strains isolated and investigated in this study.
Table S7. ANI and dDDH values should be provided. Explain what are the numbers provided in the "remarks" column.
Response: ANI and dDDH values are now included in Suppl. Table S7 (and Table S6, as well). The remarks column was removed.
SUPPL. MATERIAL & FIGS
Since there are many results, figures and tables should be carefully prepared to clearly show the results.
Please revise figures and especilly the tables since they seemed in an unfinished format. Suppl. material should be prepared with similar care as main material.
Response: Supplemental Tables, especially table S1- S9 were revised.
Different colors, fonts and highlights were used without explaining the meaning. Some info are duplicated. These makes it more confusing.
Response: Additional explanations were added to Suppl. Table S1-S7
Suppl. figs
Fig S1-S4. What does "high" similarity mean? Provide an objective criteria.
Response: “high” similarity was now replaced by “Core genome genes shared by Lysinibacillus sp. CD3-6 and Lysinibacillus sp. JNUCC-52…in Suppl. Fig. S1. The term “core genome genes” was used throughout in the other supplemental figures (Figs. S2-Fig. S4)
It is not possible to recognize RNA and MIsc features.
Response: BioCircos within the EDGAR package (Fig. 2) does not show these features.
It is not necessary to show genome names twice in the figure. The legend of the figure could be used for that.
Suppl. tables
Please explain the meaning of the colors applied in tables S1-S4.
Response: Additional explanations were added in tables S1-S4.
Table S6. Some genomes present ANI values of 100 and there is no other genome in the same cluster. What does this value mean? This table is not very clear.
Response: Tables S6 and S7 have been carefully improved. ANI and dDDH values of 100 occur, when only one single genome is present in the respective cluster and the comparison was performed against the same single genome.
Suppl. Fig S12- Please provide the meaning of the abbreviated modules in the legend. Please explain why are some modules with faded colors and others with an "X". Were these modules completely sequenced or were they in the edge og the contig?
Response: An explanation is now added to Suppl. Fig. S12.
line 711- Suppl. Fig. S21?
Fig 11. The information shown in blue is duplicated. Likewise, figures A, B and C have legends in the figure that should be shown in the legend only. Please check all your figures accordingly.
Response: The legends within the Figure (now Fig. 10) were removed.
OTHERS
line 489- Phytophthora is an oomytece, not a fungus.
Response: corrected at line 37 (introduction) and in section 3.3.2.
Reviewer 3 Report
This study was focused on 11 biocontrol bacterial strains including Brevibacillus and Lysinibacillus genus to suppress plant pathogens in Vietnamese. Full genome sequencing revealed that several of these strains represented novel genomospecies. Brevibacilli had strong biocontrol potential directed against phytopathogenic bacteria, fungi, and nematodes in vitro and in vivo. 157 natural product biosynthesis gene clusters (BGCs) were identified by genome mining, including 36 novel BGCs, not presenting in the MIBiG data bank. This study suggested a rich source of putative antimicrobial compounds, and it might serve as a new way to develop novel biocontrol agents. The manuscript is well organized and could be much better if some proofs of certain antimicrobial compounds or related gene were provided.
Major concern:
1. The result of plant growth-promoting was not shown in the abstract.
2. Too much discussion of the manuscript and not enough clarity on the main ideas make the manuscript more like a review paper than a research paper. It is suggested to simplify the results and discussion sections, and highlight the important content.
3. Were the 11 bacterial strains isolated from the same plant sample? If so, these strains may have the combined effect of inhibiting plant pathogens and promoting plant growth.
Minor mistake:
Line100, what is the full name of MES?
Line128, “2.3. Phylogeny and Genome similarity assessment” should be shown on the next row.
Line140, what do you mean by PPFMs?
Line155, what is the concentration of the bacteria?
Line 184, Arabidopsis thaliana was used to measure the effect of plant growth-promoting of bacterial strains, why not coffee and black pepper?
Line189, “bacteria (105CFU/ml)” should be “bacteria (105 CFU/mL)”
Line 177, There were three abbreviations (line 177 hrs, line188 h, 192 hr) representing hour, please keep them identical.
Line 220, Is it contaminated DNA? A pair of primers should be designed for this DNA, and then the colony as the template for PCR verification so as to eliminate the pollution.
Line 226, the LPSN showed that genus Lysinibacillus had 21 child taxa with a validly published and correct name, why only 15 were showed in this study?
Line 229, mobile genetic elements consists of gene island, prophage and CRISPR, why was only gene island mined in this study?
Line251, “CDs” should be “CDS”.
Lines 261-265, The font size here is inconsistent with that elsewhere.
Line319, “[37[.” should be “[37].”
Line 361, the name of genus and species should be italic, but not strain number.
Line 467, Poor picture quality.
Line 592, “AntiSMASH” should be “antiSMASH”.
Line 597, a space missed in “E681.Different”.
Author Response
Author's Reply to the Review Report (Reviewer 3)
Comments and Suggestions for Authors
This study was focused on 11 biocontrol bacterial strains including Brevibacillus and Lysinibacillus genus to suppress plant pathogens in Vietnamese. Full genome sequencing revealed that several of these strains represented novel genomospecies. Brevibacilli had strong biocontrol potential directed against phytopathogenic bacteria, fungi, and nematodes in vitro and in vivo. 157 natural product biosynthesis gene clusters (BGCs) were identified by genome mining, including 36 novel BGCs, not presenting in the MIBiG data bank. This study suggested a rich source of putative antimicrobial compounds, and it might serve as a new way to develop novel biocontrol agents. The manuscript is well organized and could be much better if some proofs of certain antimicrobial compounds or related gene were provided.
Major concern:
- The result of plant growth-promoting was not shown in the abstract.
Response: The result of plant growth promotion is now included in the abstract “In vitro and in vivo assays demonstrated their ability to promote plant growth, and the strong biocontrol potential of Brevibacilli directed against phytopathogenic bacteria, fungi, and nematodes.”
- Too much discussion of the manuscript and not enough clarity on the main ideas make the manuscript more like a review paper than a research paper. It is suggested to simplify the results and discussion sections, and highlight the important content.
Response: In order to avoid redundance I have merged the results and discussion section. The intention of the paper is to characterize the novel isolates (Lysinibacillus and Brevibacillus) within their exact taxonomical background. Since in my eyes the present taxonomical knowledge of these two genera is not satisfying I was forced to elucidate the taxonomical relationship in more detail, than normally necessary. I think it makes no sense to describe the genomic features and properties of the novel strains without an sufficient background of their taxonomy. I tried to work out the main idea of the paper in the conclusion section, that Brevibacilli due to their high genomic flexibility indicated by the presence of numerous extrachromosomal elements such as genomic islands and plasmids, possess an extreme potential for biocontrol of plant pathogens which is connected with a surprising diversity in producing a multitude of secondary metabolites.
- Were the 11 bacterial strains isolated from the same plant sample? If so, these strains may have the combined effect of inhibiting plant pathogens and promoting plant growth.
Response: The strains were isolated from many different sources. Geographical sites of isolation and the plant organs from them the samples were taken have been previously described (Tam, L.T.T.; Jähne, J.; Luong, P.T.; Thao, L.T.P.; Chung L.T.K.; Schneider, A.; Blumenscheit, C.; Lasch, P.; Schweder, T.; Borriss, R. Draft genome sequences of 59 endospore-forming Gram-positive bacteria associated with crop plants grown in Vietnam. Microbiol. Resour. Announc. 2020, 9,e01154-20).
Minor mistake:
Line100, what is the full name of MES?
Response: full name is 2-morpholino-ethan sulfonic acid (see section 2.1)
Line128, “2.3. Phylogeny and Genome similarity assessment” should be shown on the next row.
Response: Due to a proposal of reviewer 1 this section was merged with section 2.2
Line140, what do you mean by PPFMs?
Response: PPFM was removed in the sentence
Line155, what is the concentration of the bacteria?
Response : concentration of indicator bacteria was around 109 cells.
Line 184, Arabidopsis thaliana was used to measure the effect of plant growth-promoting of bacterial strains, why not coffee and black pepper?
Response: Arabidopsis thaliana is a widely accepted model system for plant growth promotion experiments
Line189, “bacteria (105CFU/ml)” should be “bacteria (105 CFU/mL)”
Response: corrected
Line 177, There were three abbreviations (line 177 hrs, line188 h, 192 hr) representing hour, please keep them identical.
Response: corrected to h.
Line 220, Is it contaminated DNA? A pair of primers should be designed for this DNA, and then the colony as the template for PCR verification so as to eliminate the pollution.
Response to line 220/209: It is unlikely that CP085881.1 represents contaminated DNA, but we have not investigated this in more detail. The sequence exhibits weak similarity to rep initiator proteins. Therefore, we can´t exclude the possibility that CP085881.1 represents a self-replicating element or at least a part of them.
Line 226, the LPSN showed that genus Lysinibacillus had 21 child taxa with a validly published and correct name, why only 15 were showed in this study?
Response: According to Fig. 3A there are nine validly published type strains in Lysinibacillus branch B. In the more related branch A, there are according to Suppl. Table S6 at least ten validly published species. In total, alone branch A harbors 24 species clusters, which are listed in Suppl. Table S6. They were all involved in the comparison.
Line 229, mobile genetic elements consists of gene island, prophage and CRISPR, why was only gene island mined in this study?
Line251, “CDs” should be “CDS”.
Response: corrected to CDS.
Lines 261-265, The font size here is inconsistent with that elsewhere.
Response: was corrected to 12 pt.
Line319, “[37[.” should be “[37].”
Response: corrected
Line 361, the name of genus and species should be italic, but not strain number.
Line 467, Poor picture quality.
Response: Resolution of the JPEG file is 3,16 MB
Line 592, “AntiSMASH” should be “antiSMASH”.
Response: corrected
Line 597, a space missed in “E681.Different”.
Response: E681 is the correct designation and is used throughout the manuscript.